# Magnitude of workplace violence and associated factors among healthcare professionals in East Africa: A systematic review and meta-analysis

Yeshiambaw Eshetie[1]*, Astewle Andargie Baye[1], Mengistu Ewunetu[1], Gashaw Kerebeh[2], Worku Necho Asferie[2], Demewoz Kefale[2]

1 Department of Adult Health Nursing, College of Health Science, Debre Tabor University, Debre Tabor, Ethiopia, 2 Department of Pediatrics and Child Health Nursing, College of Health Sciences, Debre Tabor University, Debre Tabor, Ethiopia

* yeshiambaweshetie@gmail.com

## Abstract

### Introduction

Workplace violence refers to violent acts that occur in the workplace against employees while they are delivering services to consumers. This phenomenon is an invasive and alarming issue affecting employees worldwide, posing both implicit and explicit threats to their health, safety and well-being. According to a World Health Organization report and study findings, 20% to 38% of healthcare workers have experienced physical violence at some point during their careers, compared to employees in other sectors. This study aimed to assess the pooled magnitude of workplace violence and to identify its associated factors among healthcare professionals in East Africa.

### Method

The study protocol was registered with PROSPERO under registration number CRD42024552266. An extensive electronic database search was conducted from August 10–31, 2024, using PubMed, Google Scholar, Web of Science, and manual Google searches. The extracted data were exported into STATA version 17 for analysis. A weighted inverse-variance random-effects model was used to calculate the pooled magnitude of workplace violence and to determine the impact of predictors on the workplace violence. Publication bias was checked by a funnel plot and Egger's test. Heterogeneity was assessed using $I^2$ statistic and Galbraith plot. Subgroup and sensitivity analyses were conducted to investigate the sources of heterogeneity.

### Results

A total of 25 studies involving 9,648 participants were included in this study. The pooled magnitude of workplace violence was 55.64% (95% CI: 48.32, 62.96; $I^2$ = 97%,

**Data availability statement:** All relevant data are within the paper and its Supporting Information files.

**Funding:** The author(s) received no specific funding for this work.

**Competing interests:** The authors have declared that no competing interests exist.

**Abbreviations:** WPV, Workplace violence; AOR, Adjusted Odds Ratio; JBI, Joanna Briggs Institute; CI, Confidence Interval; PRISMA, Preferred Reporting Item for Systematic review and Meta-analysis.

$p < 0.01$). Factors significantly associated with workplace violence included working in the emergency department (AOR = 4.3, 95% CI: 3.22, 5.39), younger age (AOR = 3.01, 95% CI: 1.42, 4.60), less work experience (AOR = 5.14, 95% CI: 2.67, 7.61), being female (AOR = 2.74, 95% CI: 1.54, 3.95), and alcohol consumption (AOR = 3.17, 95% CI: 1.52, 4.83).

## Conclusion

The magnitude of workplace violence in the region was relatively prevalent, with significantly higher odds among emergency department staff, younger healthcare professionals, those with less work experience, female professionals, and individual reporting alcohol consumption.

## Introduction

Workplace violence (WPV) refers to violent acts that occur in the workplace and are directed against employees while they are delivering services to consumers [1,2]. This phenomenon, which may occur incidentally or recurrently, is an invasive and alarming issue affecting employees worldwide, posing both implicit and explicit threats to their health, safety, and well-being. Through such unpleasant and violent acts, staff are abused, threatened, and assaulted in relation to the services they provide [3]. Workplace violence is more common in healthcare settings, which account for a quarter of all workplace violence cases [3]. According to a World Health Organization report and study findings, 20% to 38% of healthcare workers have experienced physical violence at some point during their careers, compared with approximately 26% of employees in the banking sector [4–7]. Due to significant underreporting of violent acts, the number of victims is above the figure aforementioned.

Healthcare professionals are more vulnerable to this phenomenon due to their prolonged interaction with healthcare service consumers, compared to employees in other sectors [8,9]. Distressingly, study findings have highlighted that, due to repeated exposure to violence, professionals have come to view such incidents as part of their job description [10]. Studies conducted globally have shown that the prevalence of workplace violence ranges from 19.33% to 61.9% [11,12]. Specifically, the prevalence has been reported as 62.4% in China [13], 58.4% in Saudi Arabia [14], and 63% in India [15]. In Africa, prevalence ranges from 9% to 62.3% [16,17], with country-specific figures such as 39.1% to 100% in Nigeria [18], 56% in Ethiopia [19], and 40.7% in Kenya [20]. In addition, several predictors have been identified as contributing factors to workplace violence, including substance use, work units, gender, years of experience, training on violence prevention, staffing levels, and workers' age [18,21–23].

As a result of its profound impact on the healthcare sector, the issue of workplace violence against healthcare professionals has gained increasing attention in recent years [24]. This violence not only threatens the safety of healthcare

professionals but also affects the quality of care and outcomes within healthcare organizations. Acts of workplace violence create an atmosphere of job insecurity, fear, and psychological distress, resulting in both mental and physical health problems for workers [10,25,26]. Study findings reveal that professionals who experience workplace violence are more likely to suffer from burnout, sleep disorders, turnover intentions, and an increased risk of impaired work quality [27–29].

Indeed, WPV has a damaging impact on both the quantity and quality of services provided in the health sector worldwide. This issue is particularly severe in Africa, especially in East African countries, where healthcare institutions face similar challenges across the region. Despite the importance of investigating workplace violence at the regional level, the available evidence on this phenomenon remains fragmented. There is no comprehensive study on workplace violence in East African countries, despite their similarities in resources, healthcare workforce, technology use, infrastructure, and healthcare practices. It is therefore feasible to assess the magnitude of workplace violence and its predictors at the regional level. Although patient care is a fundamental aspect of healthcare facilities, the insecurity experienced by healthcare professionals reduces both the quality and quantity of services provided, leading to poor organizational outcomes. To address the aforementioned information gaps, we conducted a systematic review and meta-analysis to assess the magnitude of workplace violence and its associated factors among healthcare professionals in East Africa.

## Methods

### Reporting and registration protocol

The results of this study were reported in accordance with the Preferred Reporting Item for Systematic Review and Meta-Analyses (PRISMA) [30] guidelines (S1 File). The protocol for this systematic review was registered with the International Prospective Register of Systematic Reviews (PROSPERO) under registration number CRD42024552266 prior to the start of data extraction.

### Search strategy and databases

An extensive electronic search was conducted using the following databases: PubMed, Google Scholar, and Web of Science. In addition, manual searches using Google were also performed. Prior to the main search, two authors (G.K. and D.K.) conducted a pilot test of the search strategy in PubMed from July 5–10, 2024. The full electronic search was then carried out from August 10–31, 2024. To include gray literature, a search for unpublished studies was also conducted using online university repositories. Expertise was involved to search complementary articles that were not rescued by searching electronic data files or reference lists. Condition (Co), Context (Co), and Population (Po) (CoCoPoP) framework was used in the study design. The relevant search terms comprised a combination of pertinent Medical Subject Headings (MeSH) and free-text keywords, with East African countries employed as a search filter. Boolean operators (AND, OR) were used to construct the search string (S2 File). For example, the search string for one database was: "workplace violence" OR "threaten" OR "aggression" AND "prevalence" OR "magnitude" AND "associated factors" OR "predictors" AND "healthcare professionals" OR "nurses" OR "doctors" OR "pharmacists" AND "East African countries".

### Inclusion and exclusion criteria

This review included all quantitative and mixed-method observational studies (focusing on the quantitative component) that were written in English, conducted among employed healthcare professionals, and reported the prevalence of workplace violence and/or at least one associated factor in East Africa. However, studies lacking abstracts and/or full-text articles, those that did not report the outcome of interest, as well as systematic reviews, meta-analyses, and qualitative studies, were excluded.

## Study selection procedures

To remove duplicate studies, all retrieved records were exported to EndNote Reference Manager, version seven. To determine the eligibility of each study, two reviewers (Y.E. and D.K.) independently screened the titles and abstracts of all studies before reviewing the full-text articles. A third reviewer (A.A.B.) was involved to resolve any disagreements between the two authors.

## Quality assessment

The quality of each primary study was assessed using the Joanna Briggs Institute (JBI) Critical Appraisal Checklist for Studies Reporting Prevalence Data [31]. The quality evaluation focused on the appropriateness of the study participants, settings, design, as well as appropriateness of measurements used to report outcomes. A primary study that achieved a score of 50% or higher on the Joanna Briggs Institute Critical Appraisal Checklist used for quality evaluation was deemed to meet high-quality standards and was included in the review. Consequently, the included studies were selected based on the quality evaluation scores, which ranged from 77.8% to 100%. Of the included primary studies, two scored seven out of nine, fifteen scored eight out of nine, and eight scored nine out of nine (Table 1).

## Data extraction

Using a standardized Microsoft Excel 2010 spreadsheet, all necessary data were extracted by two reviewers (Y.E. and A.A.B.), and cross-checked the results to ensure consistency. When inconsistencies occurred, a third reviewer (M.E.) was involved to resolve the discrepancies between the data extractors. The data extracted from the selected studies included the first author's name, publication year, study design, study area, sample size, prevalence, odds ratios, the lower and upper bounds of the odds ratios for factors significantly associated in each primary study.

## Outcome of interest

The primary outcome of this review was the pooled prevalence of workplace violence among healthcare professionals in East Africa. The associated factors of workplace violence were the second interest of this meta-analysis.

## Operational definition

In this study, workplace violence against healthcare professionals is defined as any act of physical assault, psychological abuse, verbal insult, or sexual harassment that occurs in healthcare settings and is perpetrated by patients, caregivers, coworkers, or supervisors. Studies are included if they quantitatively assess at least one form of workplace violence using a standard or clearly defined measure and report data sufficient to calculate effect size.

## Data analysis

The extracted data were exported to STATA version 17, and metadata were declared for statistical analysis. A weighted inverse-variance random-effects DerSimonian-Laird model was used to calculate the pooled prevalence of workplace violence and to assess the impact of its predictors on the outcome variable [32]. Publication bias was checked by funnel plot and Egger's test with p-value < 0.05% [33,34]. Heterogeneity among the included studies was assessed using $I^2$ statistics and the Galbraith plot [35]. The $I^2$ statistic values of 0%, 25%, 50%, and 75% represent no, low, moderate, and high heterogeneity, respectively. A p-value of less than 0.05 from the Q test was used to indicate substantial heterogeneity. A sensitivity analysis was performed to assess the influence of individual studies on the overall meta-analysis. To estimate the pooled adjusted odds ratios (AOR) of the independent factors, a forest plot was used, and the measures of association were reported with 95% confidence intervals. The adjusted odds ratio was the most commonly reported measure of

**Table 1. Quality assessment of included studies using JBI'S Critical appraisal checklist for cross-sectional studies.**

| Author (Year) | JBI'S Critical Appraisal questions | | | | | | | | | Score (100%) | Included |
|---|---|---|---|---|---|---|---|---|---|---|---|
| | Q1 | Q2 | Q3 | Q4 | Q5 | Q6 | Q7 | Q8 | Q9 | | |
| Bekalu et al (2023) | Y | Y | Y | Y | Y | Y | N | Y | Y | 88.9 | ✓ |
| W/Senbet et al(2022) | Y | Y | Y | Y | Y | N | Y | Y | Y | 88.9 | ✓ |
| Abeya et al (2018) | Y | Y | Y | Y | Y | Y | Y | Y | Y | 100 | ✓ |
| Legesse et al (2022) | Y | Y | N | Y | Y | Y | Y | Y | Y | 88.9 | ✓ |
| Anose et al (2024) | Y | Y | Y | Y | Y | Y | Y | Y | Y | 100 | ✓ |
| Likissa et al (2014) | Y | N | Y | Y | Y | Y | Y | Y | Y | 88.9 | ✓ |
| Alemu et al (2023) | Y | Y | N | Y | Y | Y | Y | Y | Y | 88.9 | ✓ |
| Abate et al (2019) | Y | N | Y | Y | Y | Y | Y | Y | Y | 88.9 | ✓ |
| Dagnaw et al (2021) | Y | Y | N | Y | Y | Y | Y | Y | Y | 88.9 | ✓ |
| Tiruneh et al (2016) | Y | Y | Y | Y | Y | Y | Y | Y | Y | 100 | ✓ |
| Mekonen et al (2020) | Y | Y | N | Y | Y | Y | Y | Y | Y | 88.9 | ✓ |
| W/Hawaryat et (2020) | Y | Y | N | Y | Y | N | Y | Y | Y | 77.8 | ✓ |
| Wubneh et al (2023) | Y | Y | Y | Y | Y | Y | Y | Y | Y | 100 | ✓ |
| Yenealem et al (2024) | Y | Y | N | Y | Y | Y | Y | Y | Y | 88.9 | ✓ |
| Yenealem et al (2019) | Y | Y | Y | Y | Y | Y | Y | Y | Y | 100 | ✓ |
| Tolera et al (2024) | Y | Y | N | Y | Y | Y | Y | Y | Y | 88.9 | ✓ |
| Fute et al (2015) | Y | N | Y | Y | Y | Y | Y | Y | Y | 88.9 | ✓ |
| Dagnaw et al (2022) | Y | Y | Y | Y | Y | Y | Y | Y | Y | 100 | ✓ |
| Kibunja et al (2021) | Y | Y | Y | Y | Y | Y | Y | Y | Y | 100 | ✓ |
| Atogo et al (2024) | Y | N | Y | Y | Y | Y | Y | Y | Y | 88.9 | ✓ |
| Musengamana et al(2022) | Y | Y | Y | Y | Y | Y | Y | Y | N | 88.9 | ✓ |
| Newman et al (2011) | Y | N | Y | Y | Y | Y | Y | Y | Y | 88.9 | ✓ |
| Elhadi et al (2022) | Y | Y | Y | Y | Y | Y | Y | Y | Y | 100 | ✓ |
| Elamin et al (2020) | Y | N | Y | Y | Y | Y | Y | Y | Y | 88.9 | ✓ |
| Gaafar et al (2022) | Y | Y | Y | Y | Y | N | Y | Y | N | 77.8 | ✓ |

**Abbreviations and detail of each questions:** Yes, N: No, Q: Question. The total quality result is calculated by counting the number of y in each row. Q1: Was the sample frame appropriate to address the target population? Q2: Were study participants sampled in an appropriate way?/Are the patients at a similar point in the course of their condition/illness? Q3: Was the sample size adequate? Q4: Were the study subjects and the setting described in detail?/Are confounding factors identified and strategies to deal with them stated? Q5: Was the data analysis conducted with sufficient coverage of the identified sample?/Are outcomes assessed using objective criteria? Q6: Were valid methods used for the identification of the condition? Q7: Was the condition measured in a standard, reliable way for all participants? Q8: Was there appropriate statistical analysis?/Were outcomes measured in a reliable way? Q9: Was the response rate adequate, and if not, was the low response rate managed appropriately.

association in the eligible primary studies, and a random-effects model was used to estimate the pooled AOR in this study. Results were presented using tables, figures, and texts.

## Results

### Study selection

An extensive search was conducted, and a total of 1,243 studies were retrieved from databases Google Scholar (510), Web of Science (200), PubMed (510), manual Google searches (13), and the university's online research repository (10). After removing duplicate studies (106) and excluding irrelevant ones based on their titles and abstracts (1,010), a total of 127 studies were selected for full-text article review. During the full-text review, 89 studies were excluded because the full-text articles did not report the outcomes of interest. Of the remaining 38 studies, 13 were excluded due to differences

in study design. Finally, 25 studies were included for data extraction to determine the pooled prevalence and associated factors of workplace violence in East Africa (Fig 1).

## Characteristics of the included studies

Finally, a total of 25 studies were deemed suitable and included in this systematic review and meta-analysis. Of these, 4 studies did not report the prevalence of workplace violence but provided information on the types and predictors of the workplace violence [36–39], whereas 14 studies reported both the prevalence and predictors of workplace violence [36,40–52], and 7 studies report only the prevalence [53–59]. Geographically, 18 studies were conducted in Ethiopia, 3 in Sudan, 2 in Kenya, and 2 in Rwanda (Table 2). Regarding tools used for outcome measurement, 21 studies used self-administered structured questionnaires, while the other four used semi-structured questionnaires and interviews (S3 File).

## Meta-analysis

### The pooled magnitude of workplace violence

A total of 25 eligible primary studies with 9,648 participants were included in the final meta-analysis. The pooled magnitude of workplace violence among healthcare professionals in East Africa was 55.64% (95% CI: 48.32, 62.96; $I^2 = 98\%$, $p < 0.01$) (Fig 2).

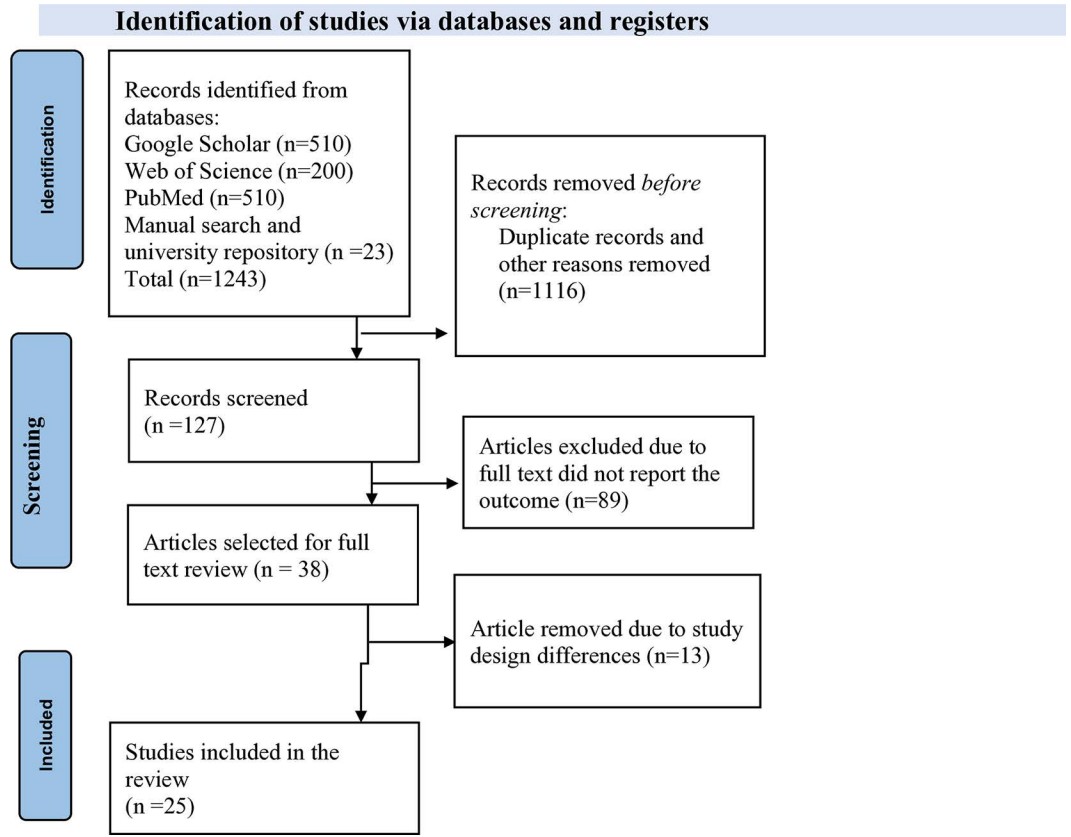

**Fig 1. PRISMA flow diagram for study selection.**

**Table 2. Characteristics of included studies for this systematic review and meta-analysis of workplace violence and associated factors in East Africa.**

| Author (Year) | Study area | Study design | Sample size | Study setting | prevalence | Population |
|---|---|---|---|---|---|---|
| Bekalu et al (2023) | Ethiopia | Cross-sectional | 534 | Institutional | 56 | Nurses |
| W/Senbet et al(2022) | Ethiopia | Cross-sectional | 339 | Institutional | N/A | Nurses |
| Abeya et al (2018) | Ethiopia | Cross-sectional | 258 | Institutional | 70.2 | All |
| Legesse et al (2022) | Ethiopia | Cross-sectional | 603 | Institutional | 64 | Nurses |
| Anose et al (2024) | Ethiopia | Cross-sectional | 415 | Institutional | 61.3 | Nurses |
| Likissa et al (2014) | Ethiopia | Cross-sectional | 203 | Institutional | 82.2 | Nurses |
| Alemu et al (2023) | Ethiopia | Cross-sectional | 381 | Institutional | 67.5 | Nurses |
| Abate et al (2019) | Ethiopia | Cross-sectional | 435 | Institutional | N/A | All |
| Dagnaw et al (2021) | Ethiopia | Cross-sectional | 503 | Institutional | 44.5 | All |
| Tiruneh et al (2016) | Ethiopia | Cross-sectional | 386 | Institutional | 26.7 | Nurses |
| Mekonen et al (2020) | Ethiopia | Cross-sectional | 217 | Institutional | N/A | All |
| W/Hawaryat et (2020) | Ethiopia | Cross-sectional | 348 | Institutional | 43.1 | Nurses |
| Wubneh et al (2023) | Ethiopia | Cross-sectional | 385 | Institutional | 51.4 | Nurses |
| Yenealem et al (2024) | Ethiopia | Cross-sectional | 339 | Institutional | 28.9 | Nurses |
| Yenealem et al (2019) | Ethiopia | Cross-sectional | 531 | Institutional | 58.2 | All |
| Tolera et al (2024) | Ethiopia | Cross-sectional | 744 | Institutional | 57.39 | All |
| Fute et al (2015) | Ethiopia | Cross-sectional | 642 | Institutional | 29.9 | Nurses |
| Dagnaw et al (2022) | Ethiopia | Cross-sectional | 385 | Institutional | N/A | All |
| Kibunja et al (2021) | Kenya | Cross-sectional | 82 | Institutional | 77.8 | Nurses |
| Atogo et al (2024) | Kenya | Cross-sectional | 184 | Institutional | 70 | Nurses |
| Musengamana et al(2022) | Rwanda | Cross-sectional | 195 | Institutional | 58.5 | Nurses |
| Newman et al (2011) | Rwanda | Cross-sectional | 297 | Institutional | 39 | All |
| Elhadi et al (2022) | Sudan | Cross-sectional | 792 | Institutional | 78.3 | All |
| Elamin et al (2020) | Sudan | Cross-sectional | 297 | Institutional | 50 | Physicians |
| Gaafar et al (2022) | Sudan | Cross-sectional | 153 | Institutional | 54.9 | All |

## Publication bias

The p-value from Egger's regression test (p<0.03), along with the asymmetric distribution of the included primary studies on the funnel plot (Fig 3a), suggests the presence of publication bias. Therefore, a trim-and-fill analysis was conducted to manage this publication bias (Fig 3b).

## Heterogeneity investigation

The Galbraith plot (Fig 4) and the $I^2$ statistic from the forest plot revealed significant heterogeneity among the included primary studies ($I^2 = 98\%$, p<0.01). Therefore, subgroup and sensitivity analyses were performed to find out the sources of this heterogeneity.

## Sensitivity analysis

We conducted a sensitivity analysis to evaluate the influence of individual primary studies on the overall meta-analysis. The forest plot indicated that each study's estimate was close to the pooled estimate, implying that no single study had a substantial impact. Therefore, the overall outcome of the meta-analysis is not influenced by any single primary study (Fig 5).

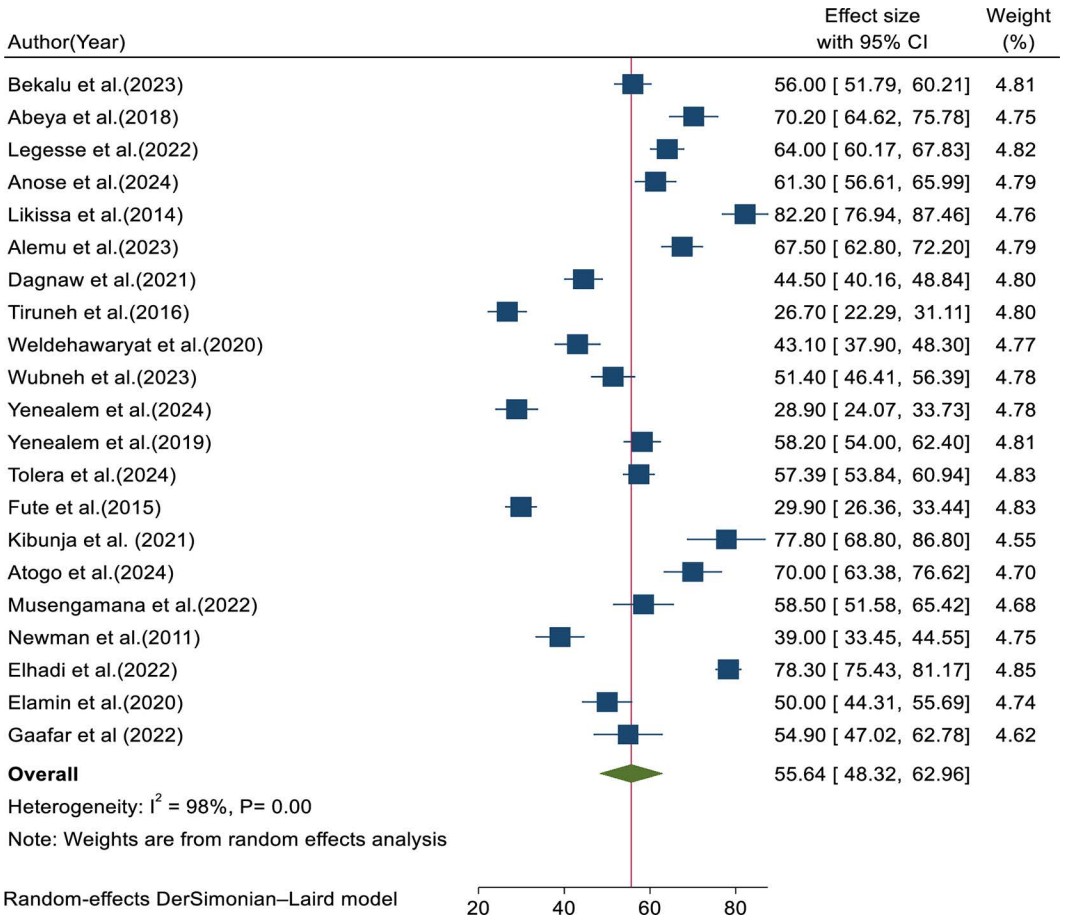

| Author(Year) | Effect size with 95% CI | Weight (%) |
|---|---|---|
| Bekalu et al.(2023) | 56.00 [ 51.79, 60.21] | 4.81 |
| Abeya et al.(2018) | 70.20 [ 64.62, 75.78] | 4.75 |
| Legesse et al.(2022) | 64.00 [ 60.17, 67.83] | 4.82 |
| Anose et al.(2024) | 61.30 [ 56.61, 65.99] | 4.79 |
| Likissa et al.(2014) | 82.20 [ 76.94, 87.46] | 4.76 |
| Alemu et al.(2023) | 67.50 [ 62.80, 72.20] | 4.79 |
| Dagnaw et al.(2021) | 44.50 [ 40.16, 48.84] | 4.80 |
| Tiruneh et al.(2016) | 26.70 [ 22.29, 31.11] | 4.80 |
| Weldehawaryat et al.(2020) | 43.10 [ 37.90, 48.30] | 4.77 |
| Wubneh et al.(2023) | 51.40 [ 46.41, 56.39] | 4.78 |
| Yenealem et al.(2024) | 28.90 [ 24.07, 33.73] | 4.78 |
| Yenealem et al.(2019) | 58.20 [ 54.00, 62.40] | 4.81 |
| Tolera et al.(2024) | 57.39 [ 53.84, 60.94] | 4.83 |
| Fute et al.(2015) | 29.90 [ 26.36, 33.44] | 4.83 |
| Kibunja et al. (2021) | 77.80 [ 68.80, 86.80] | 4.55 |
| Atogo et al.(2024) | 70.00 [ 63.38, 76.62] | 4.70 |
| Musengamana et al.(2022) | 58.50 [ 51.58, 65.42] | 4.68 |
| Newman et al.(2011) | 39.00 [ 33.45, 44.55] | 4.75 |
| Elhadi et al.(2022) | 78.30 [ 75.43, 81.17] | 4.85 |
| Elamin et al.(2020) | 50.00 [ 44.31, 55.69] | 4.74 |
| Gaafar et al (2022) | 54.90 [ 47.02, 62.78] | 4.62 |
| **Overall** | 55.64 [ 48.32, 62.96] | |

Heterogeneity: $I^2$ = 98%, P= 0.00

Note: Weights are from random effects analysis

Random-effects DerSimonian–Laird model

**Fig 2. Forest plot for pooled magnitude of workplace violence.**

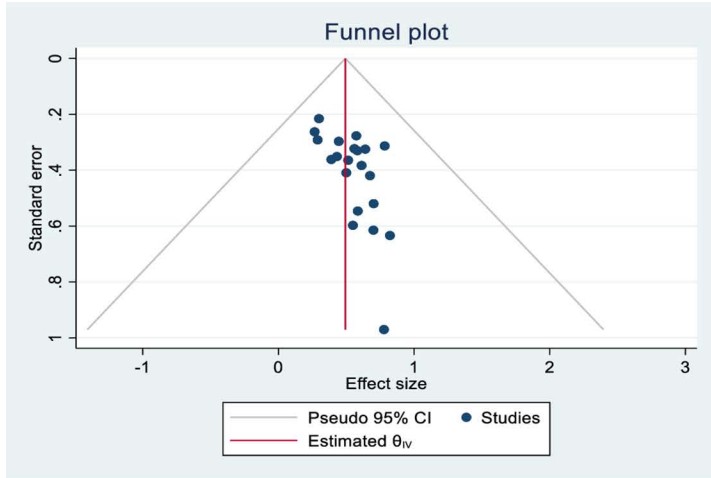
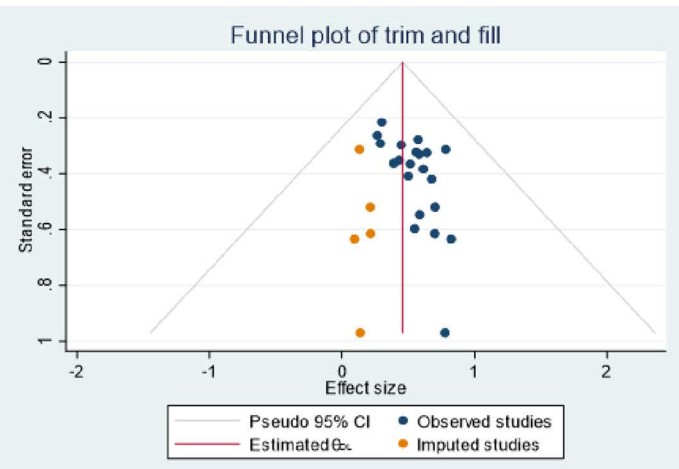

**Fig 3. Funnel plot before adjustment (3a) and after adjustment (3b).**

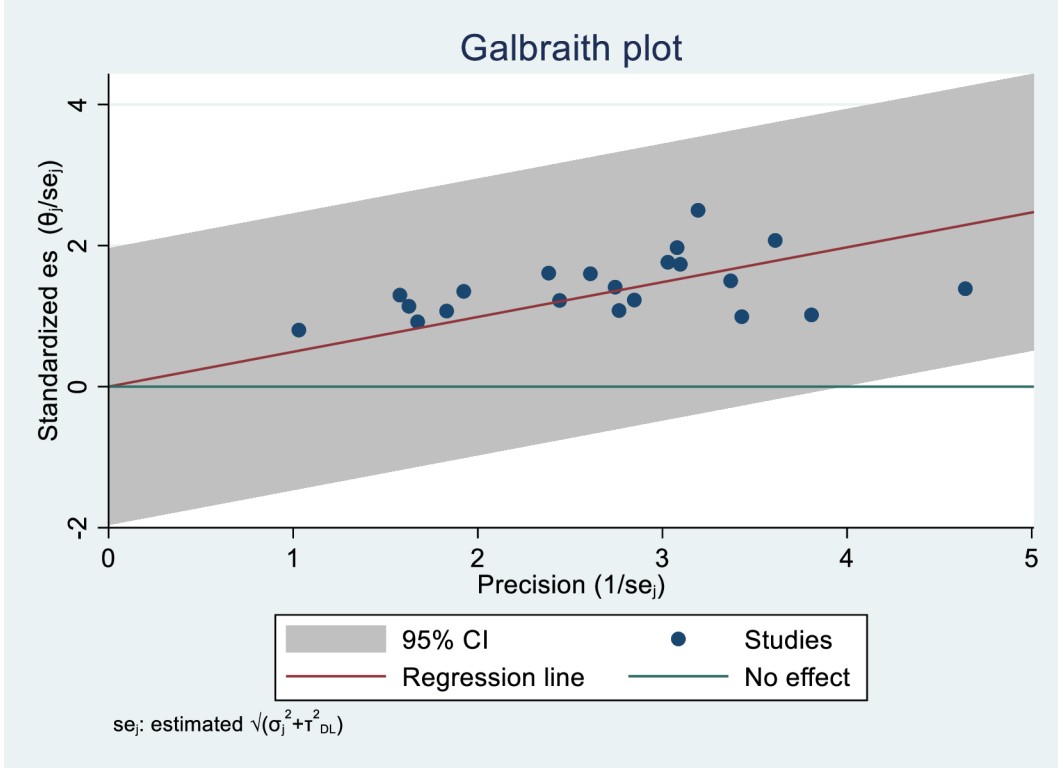

**Fig 4. Galbraith plot for heterogeneity investigation.**

## Subgroup analyses

Subgroup analysis was performed based on year of publication, sample size, and study areas for research conducted in East African countries. Based on the subgroup analysis, the pooled prevalence of workplace violence in studies published in 2022 or later was 58.96% (95% CI: 50.86, 67.07, $I^2=97.16\%$, p=0.00), showing slightly lower heterogeneity compared to studies published before 2022, which reported a prevalence of 52.02% (95% CI: 40.48, 60.56, $I^2=98\%$, p=0.00) (Fig 6).

In terms of study location, studies from Ethiopia showed the highest heterogeneity, followed by Sudan, Rwanda, and Kenya, with $I^2$ values of 98%, 97.86%, 94.62%, and 46.62%, respectively (Fig 7).

According to the subgroup analysis based on sample size, studies with an average sample size of 386 or more exhibited higher heterogeneity ($I^2=98\%$) compared to studies with less than the average sample size ($I^2=97.04\%$) (Fig 8).

The $I^2$ statistic from the subgroup analysis by study region suggests that studies from Ethiopia, Rwanda, and Sudan may be contributing to the observed heterogeneity. Although we conducted a meta-regression to quantify regional variability, the result was not statistically significant (p=0.39). This may be attributed to the limited number of studies within each regional group and warrants a cautious interpretation of the results.

## Factors associated with workplace violence

A meta-analysis was conducted to determine the pooled predictors of workplace violence among health professionals. Working in the emergency department, younger age, inadequate staffing, work experience, sex of healthcare professionals, and alcohol consumption were statistically significant factors associated with the outcome variable in the included primary studies. In this review, nine studies reported that working in the emergency department was significantly associated

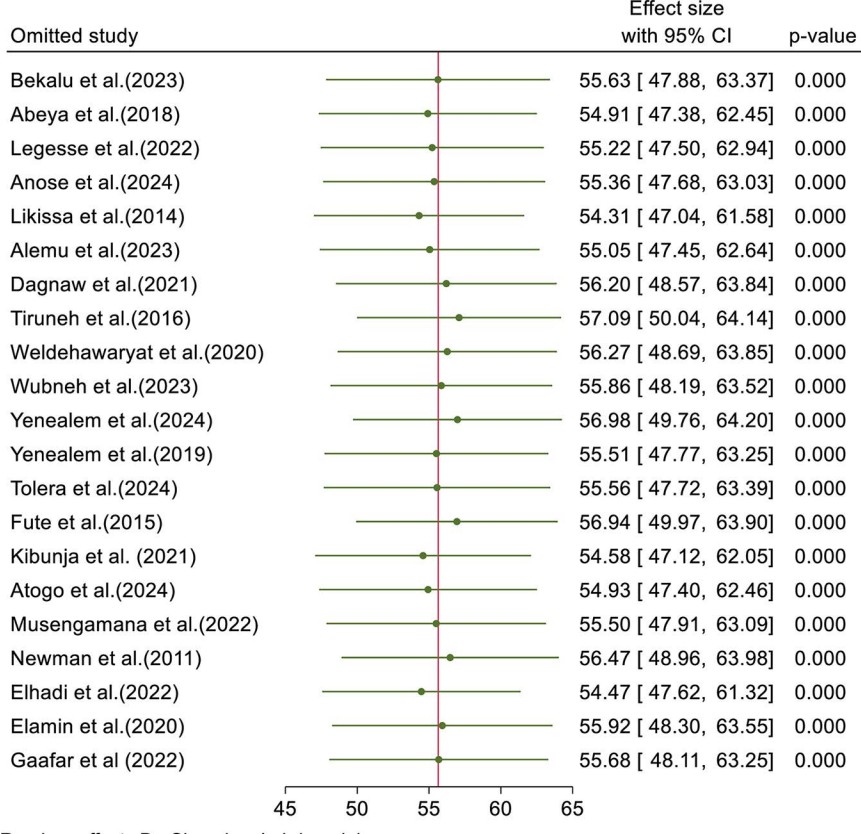

| Omitted study | | Effect size with 95% CI | p-value |
|---|---|---|---|
| Bekalu et al.(2023) | | 55.63 [ 47.88, 63.37] | 0.000 |
| Abeya et al.(2018) | | 54.91 [ 47.38, 62.45] | 0.000 |
| Legesse et al.(2022) | | 55.22 [ 47.50, 62.94] | 0.000 |
| Anose et al.(2024) | | 55.36 [ 47.68, 63.03] | 0.000 |
| Likissa et al.(2014) | | 54.31 [ 47.04, 61.58] | 0.000 |
| Alemu et al.(2023) | | 55.05 [ 47.45, 62.64] | 0.000 |
| Dagnaw et al.(2021) | | 56.20 [ 48.57, 63.84] | 0.000 |
| Tiruneh et al.(2016) | | 57.09 [ 50.04, 64.14] | 0.000 |
| Weldehawaryat et al.(2020) | | 56.27 [ 48.69, 63.85] | 0.000 |
| Wubneh et al.(2023) | | 55.86 [ 48.19, 63.52] | 0.000 |
| Yenealem et al.(2024) | | 56.98 [ 49.76, 64.20] | 0.000 |
| Yenealem et al.(2019) | | 55.51 [ 47.77, 63.25] | 0.000 |
| Tolera et al.(2024) | | 55.56 [ 47.72, 63.39] | 0.000 |
| Fute et al.(2015) | | 56.94 [ 49.97, 63.90] | 0.000 |
| Kibunja et al. (2021) | | 54.58 [ 47.12, 62.05] | 0.000 |
| Atogo et al.(2024) | | 54.93 [ 47.40, 62.46] | 0.000 |
| Musengamana et al.(2022) | | 55.50 [ 47.91, 63.09] | 0.000 |
| Newman et al.(2011) | | 56.47 [ 48.96, 63.98] | 0.000 |
| Elhadi et al.(2022) | | 54.47 [ 47.62, 61.32] | 0.000 |
| Elamin et al.(2020) | | 55.92 [ 48.30, 63.55] | 0.000 |
| Gaafar et al (2022) | | 55.68 [ 48.11, 63.25] | 0.000 |

Random-effects DerSimonian–Laird model

**Fig 5. Forest plot for sensitivity analysis.**

with health professionals' workplace violence. The pooled adjusted odds ratio for healthcare professionals working in emergency department was 4.3 (95% CI: 3.22, 5.39, $I^2$=93.71%, p=0.00) (Fig 9).

Younger age among healthcare professionals was significantly associated with the outcome variable in seven included studies, with a pooled adjusted odds ratio of 3.01 (95% CI: 1.42, 4.60, $I^2$=98.41%, p=0.00) (Fig 10).

Another predictor of workplace violence, identified in six of the included primary studies, was less work experience. The pooled adjusted odds ratio for healthcare professionals with less experience was 5.14 (95% CI: 2.67, 7.61, $I^2$=97%, p<0.001) (Table 3). Similarly, inadequate staffing was found to be statistically associated with workplace violence in four of the included studies, with a pooled adjusted odds ratio of 2.44 (95% CI: 0.90, 3.97, $I^2$=93%, p<0.001). Likewise, being a female healthcare professional was identified as a predictor of workplace violence in five of the included primary studies, with a pooled adjusted odds ratio of 2.74 (95% CI: 1.54, 3.95, $I^2$=96.88%, p<0.01). Finally, alcohol consumption was statistically associated with the outcome variable in three of the included studies, with a pooled adjusted odds ratio of 3.17 (95% CI: 1.52, 4.83, $I^2$=94.21%, p<0.001) for healthcare professionals who consumed alcohol (see table 3).

## Discussion

Although the issue of WPV against healthcare professionals has gained significant attention in developed countries, it remains inadequately assessed in East Africa. To the best of our knowledge, this is the first comprehensive, region-wide systematic review and meta-analysis conducted on this issue. The pooled magnitude of workplace violence was 55.64% (95% CI: 48.32, 62.96; $I^2$=98%, p<0.01), which is higher than the findings reported in studies from Nepal (45.5%) [60],

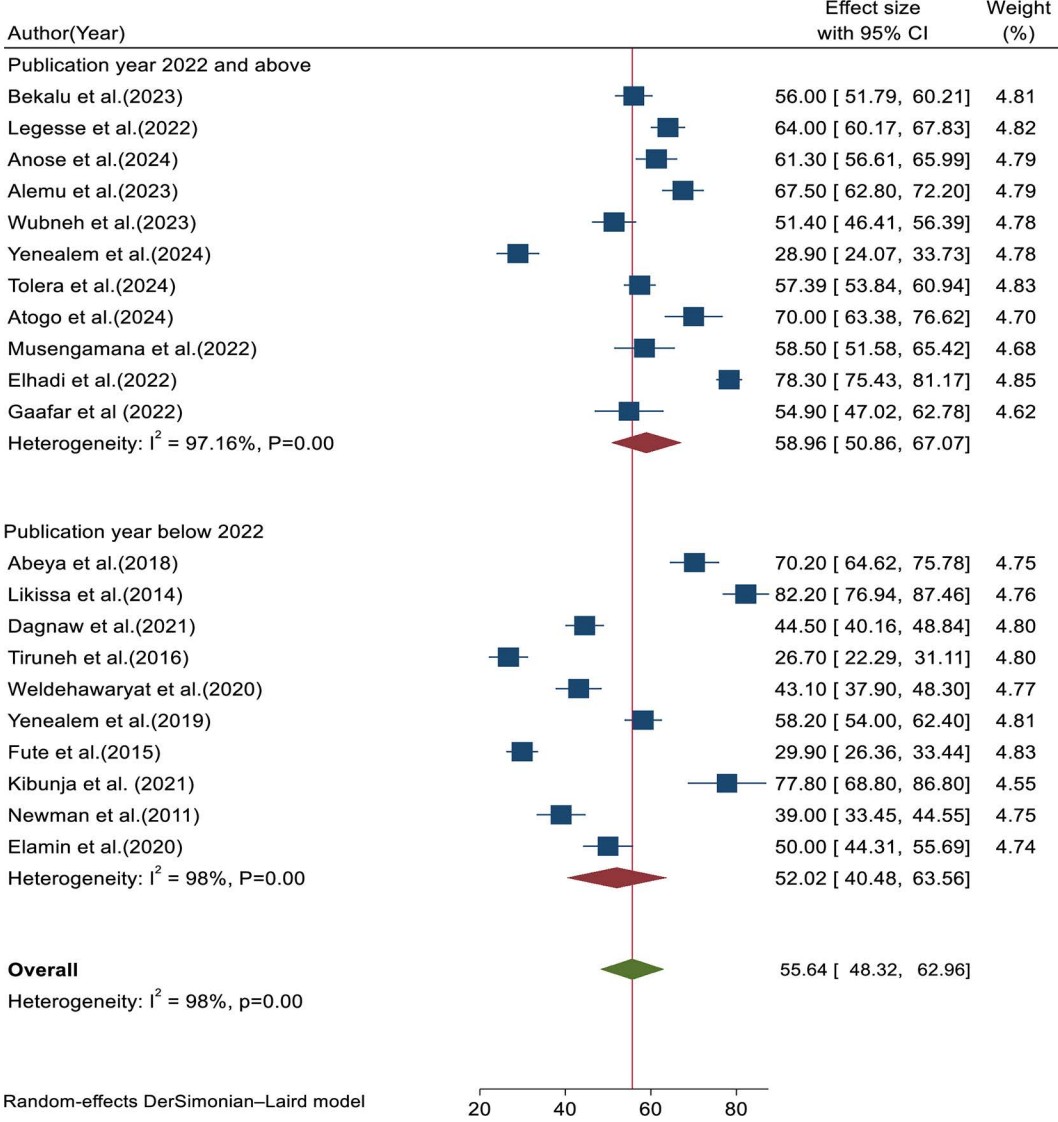

**Fig 6. Forest plot for subgroup analysis based on year of publication.**

and France (45%) [61]. A possible explanation for this disparity is that these countries have more advanced systems for reporting, investigating, protecting, and ensuring security, which help safeguard healthcare professionals as citizen by holding offenders accountable [62]. Additionally, variation in study settings, research design, and sample sizes may have contributed to the observed differences [63].

However, the magnitude found in this study is lower than that reported in studies from Africa (62.3%) [64], China (62.4%) [13], India (63%) [15], Saudi Arabia (58.4%) [14], and Bangladesh (77%) [65]. The difference from the African study may be due to its exclusive focus on nurses, a group that may have more frequent contact with patients and consequently, greater exposure to violence. The variation from the study conducted in Bangladesh may be attributed to its reliance on a single primary study rather than a pooled estimate of workplace violence. However, the results of meta-analyses reported from China, India, and Saudi Arabia may reflect findings from individual countries rather than providing data from a broader regional perspective, as the current study does [66].

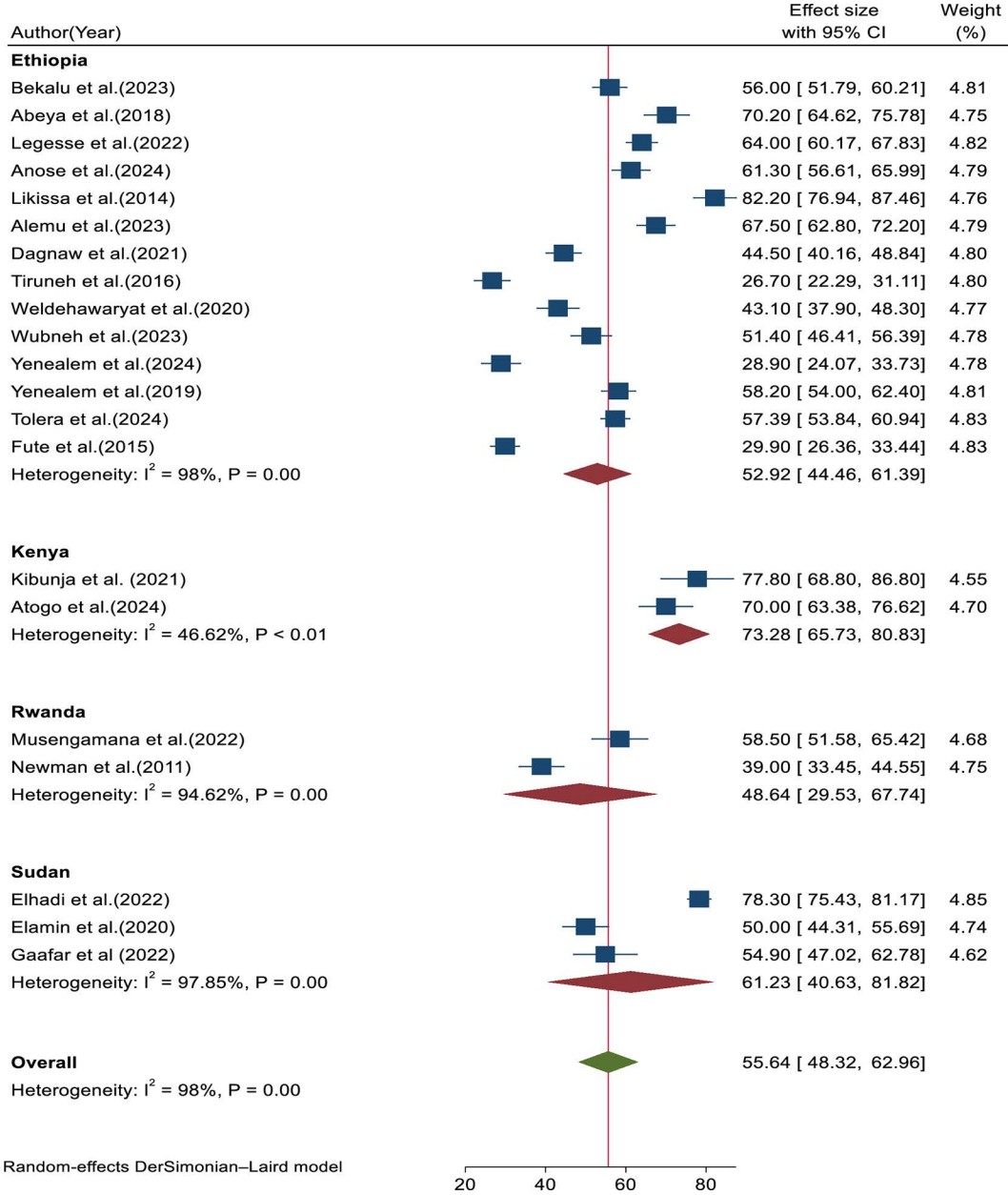

**Fig 7. Forest plot for subgroup analysis based on study location.**

In addition, the findings of this study revealed that healthcare professionals working in the emergency department were more likely to experience workplace violence compared to those working in other departments. Therefore, it is imperative to give special attention to security and foster a conducive working environment in the emergency department [67,68]. The result of this study is in line with findings of previous studies [15,69–72]. This could be due to the high patient flow in emergency department, which result in unmet needs among healthcare consumers [73]. Moreover, the emergency department is often the first place, where patients learn about their distressing diagnosis and first engage with healthcare professionals [5,74]. Similarly, the findings of this study show that younger healthcare professionals were more likely to

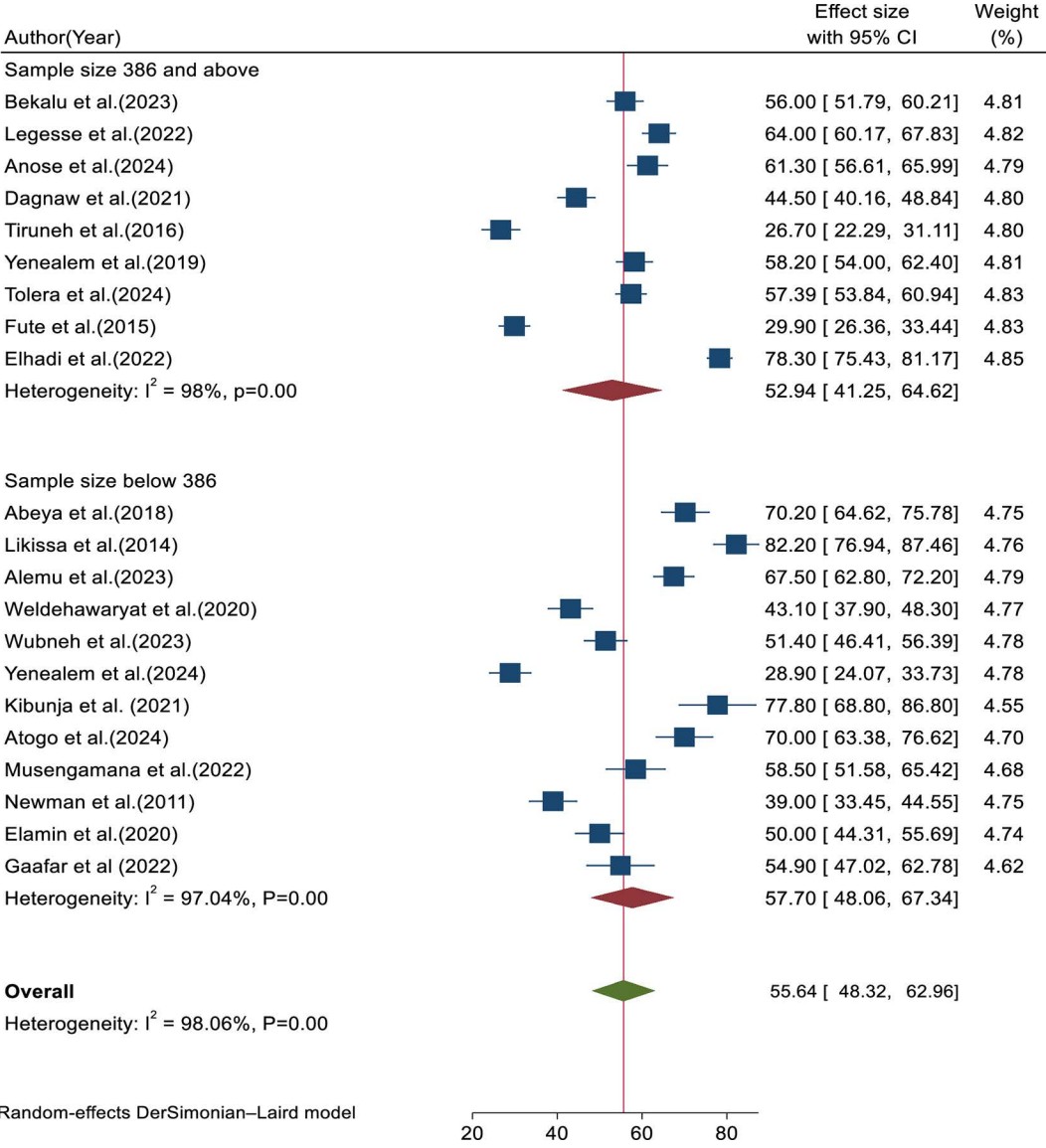

**Fig 8. Forest plot for subgroup analysis based on median sample size.**

experience workplace violence compared to their older counterparts. Hence, the assignments of healthcare professionals to each ward should include individuals of all age groups [40]. This finding is consistent with the findings of other studies [15,18,37,47,69,70]. This might be because younger healthcare professionals lack the experience and skills needed to handle and resolve disputes with healthcare service seekers [44].

Likewise, this study revealed that healthcare professionals with less work experience were more likely to encounter workplace violence than those with work experience. This finding is congruent with results of previous studies [15,69,75]. This could be attributed to the fact that new employees may possess theoretical knowledge but lack the practical skills required for certain tasks [76]. Moreover, they often face challenges in solving patients' complain regarding the services they provided [77]. Alcohol consumption is the other identified predictor of workplace violence in this study. Healthcare professionals who consume alcohol were more likely to experience workplace violence compared to those who do not.

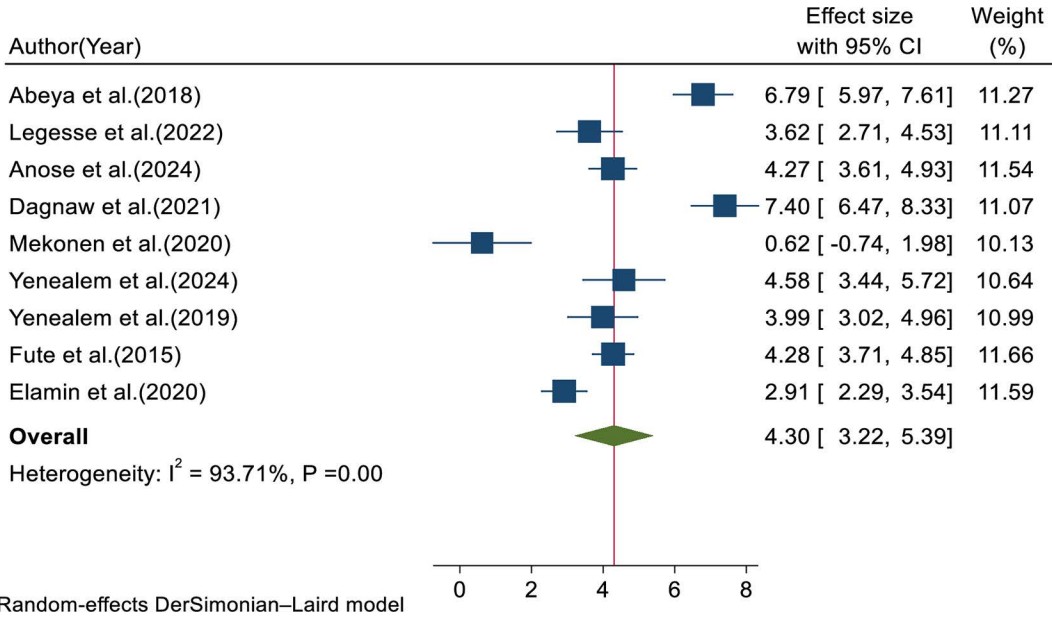

**Fig 9. Forest plot for pooled adjusted odds ratio of emergency department.**

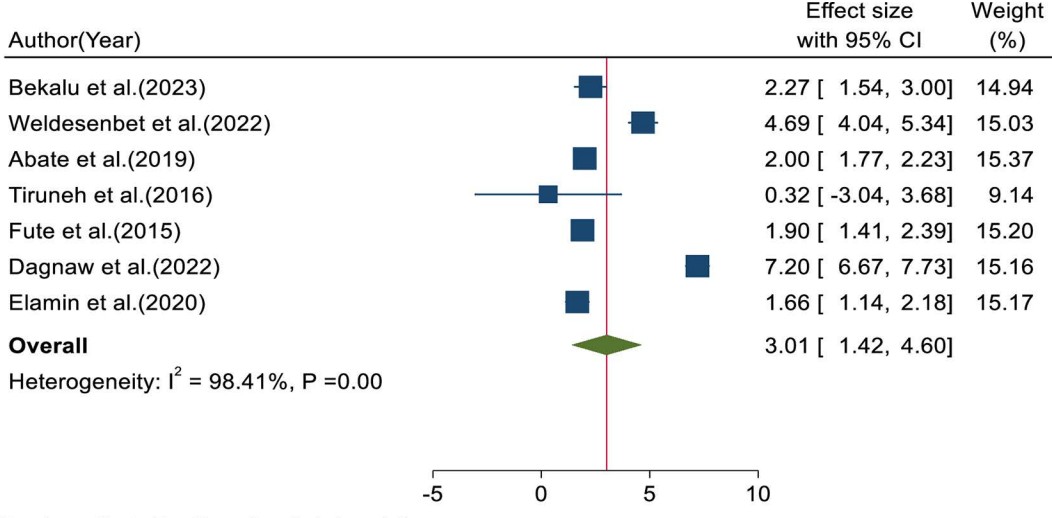

**Fig 10. Forest plot for pooled adjusted odds ratio of younger age.**

The result is in agreement with previous findings of studies [17,47]. Substances with stimulant effect can compromise professionals' ability to provide optimal patient care and tolerance, potentially leading to inappropriate reactions toward patients and their families [61,78]. Therefore, healthcare institutions should implement strict measures regarding health-care professionals who use these substances in workplace [79].

Although inadequate staffing was consistently identified as a predictor of workplace violence in the included primary studies, the pooled result of this meta-analysis revealed a statistically non-significant association between the two.

**Table 3. Showing statistically significant factors associated with workplace violence among healthcare professionals in East Africa.**

| Associated factors | Author (Year) | Pooled adjusted odds ratio with a (95% CI) | I² | P value |
|---|---|---|---|---|
| Less work experience | 1, Abeya et al (2018) | 5.14 (2.67, 7.61) | 97% | < 0.001 |
| | 2, Abate et al (2019) | | | |
| | 3, Dagnaw et al(2021) | | | |
| | 4, Yenealem et al (2019) | | | |
| | 5, Fute et al (2015) | | | |
| | 6, Dagnaw et al (2022) | | | |
| Being female healthcare professionals | 1, Bekalu et al (2023) | 2.74 (1.54, 3.95) | 96.88% | < 0.001 |
| | 2, Dagnaw et al (2021) | | | |
| | 3, Mekonen et al (2020) | | | |
| | 4, W/Hawaryat et al (2020) | | | |
| | 5, Fute et al (2015) | | | |
| Alcohol consumption | 1, Bekalu et al (2023) | 3.17 (1.52, 4.83) | 94.21% | < 0.001 |
| | 2, Weldesenbet et al (2022) | | | |
| | 3, W/Hawaryat et al (2020) | | | |

NB: I² = Heterogeneity, CI = Confidence Interval.

The non-significant association observed in the pooled result may be attributed to differences in the measurement of staffing levels, variability in the adjustment for confounders, and the presence of heterogeneous confidence intervals across the included primary studies. Future research should explore the role of staffing in workplace violence against healthcare professionals by developing a standardized assessment tool. This could enhance understanding of victims' experiences and inform targeted interventions. Finally, this study found that female healthcare professionals were more likely to experience workplace violence compared to their male counterparts. The result is congruent with the findings of previous studies [47,80]. This may be attributed to the fact that female healthcare professionals are exposed to violence not only because of the nature of the services they provide, but also due to gender-related factors, including sexual harassment [81–83].

## Limitation and strength

The absence of studies in all included countries was the prominent challenge of this study. Additionally, only a small number of studies were included due to the limited availability of published studies in each nation. In some of included studies, information on key variables, was also limited. Despite our efforts to minimize the risk of bias, there is a chance that bias may have contributed to an overestimation of workplace violence magnitude and its predictors.

## Conclusion and recommendation

The magnitude of workplace violence in the region was relatively high, posing a significant threat to the safety of healthcare professionals. This persistent sense of insecurity negatively impact the quality of healthcare services and the overall performance of institutions. Factors such as working in the emergency department, younger age, less work experience, sex of healthcare professionals, and alcohol consumption were identified as predictors of workplace violence. Based on these findings, we recommend that healthcare managers take measures to protect their employees by enhancing security in the emergency departments, considering work experience during clinical rotations, and taking into account professionals' age, and sex when assigning staff to different units.

## Supporting information

**S1 File. PRISMA 2020 reporting checklist.**
(DOCX)

**S2 File. Searching approach.**
(DOCX)

**S3 File. Titles and outcome measurement tools for included studies.**
(DOCX)

## Author contributions

**Conceptualization:** Yeshiambaw Eshetie.

**Data curation:** Yeshiambaw Eshetie, Demewoz Kefale.

**Formal analysis:** Astewle Andargie Baye, Gashaw Kerebeh, Demewoz Kefale.

**Funding acquisition:** Worku Necho Asferie.

**Investigation:** Yeshiambaw Eshetie, Astewle Andargie Baye, Gashaw Kerebeh, Worku Necho Asferie.

**Methodology:** Mengistu Ewunetu.

**Project administration:** Astewle Andargie Baye, Mengistu Ewunetu, Demewoz Kefale.

**Resources:** Yeshiambaw Eshetie, Gashaw Kerebeh.

**Software:** Astewle Andargie Baye, Worku Necho Asferie.

**Supervision:** Worku Necho Asferie.

**Validation:** Yeshiambaw Eshetie, Gashaw Kerebeh.

**Visualization:** Astewle Andargie Baye, Gashaw Kerebeh.

**Writing – original draft:** Mengistu Ewunetu.

**Writing – review & editing:** Yeshiambaw Eshetie, Mengistu Ewunetu.

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
