## [Decision Letter · Decision Letter 0]

13 Jun 2025

Dear Dr. Eshetie,

Thank you for submitting your manuscript to PLOS ONE. After careful consideration, we feel that it has merit but does not fully meet PLOS ONE’s publication criteria as it currently stands. Therefore, we invite you to submit a revised version of the manuscript that addresses the points raised during the review process.

We look forward to receiving your revised manuscript.

Kind regards,

De-Chih Lee, Ph.D.

Academic Editor

PLOS ONE

Journal Requirements:

3. Please include a copy of Table 1-2 which you refer to in your text on page 13 and 14.

Additional Editor Comments:

Please make major revisions based on the reviewer's review.

Reviewers' comments:

Reviewer's Responses to Questions

**Comments to the Author**

1. Is the manuscript technically sound, and do the data support the conclusions?

Reviewer #1: Yes

Reviewer #2: Yes

Reviewer #3: Partly

Reviewer #4: No

2. Has the statistical analysis been performed appropriately and rigorously?

Reviewer #1: Yes

Reviewer #2: Yes

Reviewer #3: Yes

Reviewer #4: No

3. Have the authors made all data underlying the findings in their manuscript fully available?

Reviewer #1: Yes

Reviewer #2: Yes

Reviewer #3: Yes

Reviewer #4: Yes

4. Is the manuscript presented in an intelligible fashion and written in standard English?

Reviewer #1: No

Reviewer #2: Yes

Reviewer #3: No

Reviewer #4: No

Reviewer #1: Manuscript Number: PONE-D-24-46926

Manuscript Title: Magnitude of workplace violence and associated factors among health professionals in East Africa. Systematic review and meta-analysis

Congratulations dear authors on your scholarly work based on a priori protocol registered in PROSPERO; you have brought an important study problem with good findings that have public health importance in optimizing health care professional safety and well-being in work place. However, there few methodological issues that I want you to address before considering the manuscript for publication.

Comments

Abstract

Introduction: This section needs to tell the reader about overview of burden of the problem, gap justification/ reason/aim of study.

Methods: better to remove the aim of the study from this part and add to introduction section. Please add sub group analysis, sensitivity analysis, study period etc…

Result: you need to reduce, make it short and precise (instead of writing associated variable and AOR separately write like (Emergency department (95%CI: 3.22, 5.39; I2 =93.71%, p = 0.00).Please write total study participants and included studies.

Conclusion: “The magnitude of workplace violence in the region was high” do you have benchmark to say high?

Your conclusions need to be based on your finding. I recommend you to place your Prospero registration here (Abstract)

Introduction (main): I feel it is well stated and organized except first write the full term before abbreviation, see your grammatical and language usages. In addition, you better to state the known factors/ variables, the attempted solution taken and planed strategies.

Methods: do you think that PIOC mnemonic is suitable for this review? Please revise it.

Please operationalize “workplace violence”

Result: Under “Characteristics of the included studies” be consistent write all in words or number (4, eighteen, 7)

It is better to narrate your result in intelligible, short and conclusive manner (revise it)

Discussion: I recommend you to revise your discussion section critically.

• Please use reference for your justification,

• Make it short and clear to transfer your message well to the reader

• There are some sentences which placed unwisely like “As primary articles revealed that……?

• The flow of idea is not smart

• Please use paragraph for distinct idea

• Generally it needs deep revision

Your conclusion should base your result

General comments

There are several typological and grammar usage errors that need extensive proof reading for revisions.

Don’t use the conjunction ‘or’ to illustrate more, just one phrase or sentence is enough

Good luck!!

Reviewer #2: Thank you for the opportunity to review this manuscript. This meta-analysis estimates the prevalence of life-long workplace violence among healthcare workers in East Africa, understanding workplace violence as having suffered at least once an aggression from a client or user. The meta-analysis is generally well perfomed and contains thorough analyses. I have one major comment which I hope will help improve this manuscript.

The meta-analysis 25 studies in its main analysis and obtains a robust estimation of the workplace violence prevalence. However, when it comes to analyzing moderator variables, the manuscript's approach is suboptimal in the sense that it does not take all of the possible studies. For instance, let me take the example of the analysis of sex as a potential variable. The manuscript only take those studies where men and women were compared and use the OR as a size effect. However, it is also possible to take the proportion of men and women in every study and perform a meta-regression, using the prevalence as the effect size, and thus taking all of the studies instead of only some of them. Same logic could be applied to age, and years of experience, and perhaps other moderating variables as well.

The manuscript also mentions that the autohrs searched on databases like PubMed, Google Scholar, and Google. Is this the complete list of databases?

As a minor comment, regarding abbreviations and their use, it is customary that, the first time a term appears, both the expanded term and its abbreviation are mentioned, and in subsequent apprearances only the abbreviation is used. I suggest the manuscript to abide to this custom. For instance, the first time the term appears is in the first sentence: there I would say "Workplace violence (WPV) is a violent...", and use only the abbreviation WPV afterwards. Same rule may be applied to the abbreviation AOR.

Last, in the last paragraph of the Methods, it is said that p-value "<0.05% was used". Do you mean a significance threshold of 0.05?

Reviewer #3: In a meta-analysis, this paper deals with the questions of how high workplace violence is in East African countries and which predictors were found. It is a methodologically sound study. This and the results found are the strength of the work. However, I do see a some weaknesses in the work. Apart from a few points at the end, the introduction is good, but a chapter with related studies is missing. I am thinking in particular of systematic reviews and meta-analyses on the issues relating to East Africa, but also worldwide for comparison. For this reason, the discussion also suffers because there is no holistic reference to that existing literature. The methods section is complete in itself, but some more precise explanations are missing, and it could generally be more detailed. In my view, a very important question is why 70% of the studies were excluded due to unavailability before the full text review. It is very important to assess whether these could have an influence on the results.

I think this study is an important contribution to the documentation and prevention of workplace violence. I hope that you can revise this meta-analysis so that it can be published.

General comments

• In chapter 3 (methods), 3.1 Data analysis follows 2.4 Data extraction. I suppose it should be 2.5 Data analysis.

You have no chapter 3. After chapter 3. Methods chapter 4. Results follows, and the 4. Meta-Analysis.

• Please, let proofread your paper. There as some grammatical errors and stylistic issues.

Abstract

The abstract is comprehensible and contains the relevant information.

• There are mentioned findings of a WHO study on WPV that are not mentioned in the introduction.

• I recommend to mention all databases searched not “like PubMed, …”.

• Egger’s test (not egger’s test)

• AOR is used in the abstract. I would write out the abbreviation in the abstract and introduce the abbreviation at first use later in the paper.

• Do not reference to figures in the abstract (“fig.12”).

• The second part of the results (in the abstract) “Emergency room AOR was …” is not written in complete sentences.

Introduction

I see some issues in the last section of the introduction.

You write that WPV is “predominantly dangerous in Africa, especially in East African countries”. Why is that the case. You should explain this.

“they are almost similar in resource, health staff and technology usage, infrastructures, health care practice and environment.” -> similar to which countries.

You write that there is no comprehensive study on WPV in East Africa and that despite the importance there are few information. But then you write “It is therefore feasible to determine the magnitude workplace violence and its predictors in the region level.”

That is not logic to me. If there are few studies, few information, how can you conduct a meta-analysis?

• There are phrases that are not easy to understand for me (e.g. “having almost similar healthcare institutions and challenges in the region of this”)

• Introduce the abbreviation of workplace violence at it’s first use; not further down in the introduction.

• “the rest employees” -> the other employees

• Several times in the paper you mention to conduct “a systematic review and a meta-analysis”. I would only refer to meta-analysis. A meta-analysis is a kind of systematic review, but I would not mention both.

Related works

I miss a chapter with related works, especially systematic reviews and meta-analyses. You might use these studies to show the actual state of the art and make first conclusions about WPV worldwide, in Africa and East Africa. Then you can bring them and their results in relation to the results of your paper in the discussion section. I listed some systematic reviews and meta-analysis further below.

Methods

In the methods section all necessary information is there, even if mostly in a rather short way. I recommend integrating the subchapters 4.1, 4.2 and 4.3 into the methods section.

But there are some points to clear.

• mObjective: “to determine the overall pooled magnitude of workplace violence and identify associated factors among healthcare professionals in East Africa”

But in the next phrase you write “to evaluate the category and predictors of workplace violence in this area”. What do you mean with category? That seems to be a new element.

• PRISMA: I would very brief explain what these guidelines are for.

• In subchapter 2.1, I do not find the description of the study design. There is the objective and the reference to the PRISMA guidelines and the register of systematic reviews.

• 2.1.1: The details of the search strategy are not clear to me. Did you mention all databases or only a part of them (“like PubMed, …”)? Who was conducted the search for grey literature? In what databases? Did you find grey literature? How was “search complementary articles” done, by whom?

• Please explain shortly what the PIOC pursuing design is.

• “The explanation approach and quantity of rescued article to database has been given.» Where did you do this?

I recommend integrating the chapters below into the methods section, where there is the reference to fig. 1.

• Concerning data extraction, you write that you did extract “odds ratio of significantly associated factors”. Did you leave out non-significant factors? That would bias results. I suppose I am misunderstanding something.

• “The associated factors of workplace violence were also the second interest of this review”

correct: “The associated factors of workplace violence were the second interest of this meta-analysis.”

• Egger’s test (not egger’s test)

• “I2 value of 0, 25%, 50 and 75%” -> “I2 value of 0%, 25%, 50% and 75%”

• Figure 1: What are other reasons for exclusion before screening

I am a bit alarmed to read that 89 of 127 studies are excluded due to unavailability of a full text. What is the unavailability? Are they not open source? Did you find only a reference but no functioning link or they seem not to exist? This seems important to me.

Whenever possible you should select these studies for full text review. If this is not possible you should estimate with the existing information (title, abstract) if these papers would change your findings.

Results

As mentioned above I recommend integrating chapter 4.1 to 4.3 into the method section.

• 4.3: the first sentence is a repetition (methods section)

Meta-analysis

When placing the chapters 4.1-4.3 into to method section, I would rename chapter 4. Meta-analysis in 4. Results.

All in all, the results are presented clear and understandable. Even if an editorial question, I would reflect about the integration of the figures within the text. Thirteen figures are a lot and most of them might be rather large compared to their contained information.

• When writing about the analysed papers I would use the word studies (not articles).

Discussion

Principally, the discussion is ok. But as there are only view related studies are cited in the introduction, I miss a broader discussion in relation to existing literature (especially systematic reviews and meta-analyses).

• I would integrate numbers (results of other studies) into the discussion, e.g. “which is higher than the findings of studies conducted in Nepal[10], in France[43].» I would add there the lower values of the referred studies.

• Alcohol and substance consumption. Reflection about cause and effect: Alcohol or substance consumption could be a consequence of WPV not a cause.

• Women face more WPV. What about sexual harassment? That’s not WPV per se, evtl. one version of it. Could it be that this is one reason for the higher values.

• This finding is supported with the result of other studies[45]. I would mention that [45] is a meta-analysis; else it seems strange to mention other studies and refer only to one.

• “This insecurity feeling of professionals directly affect the healthcare service provided and the outcome of institutions. .” -> Two points at the end of the sentence.

Meta-analyses and systematic reviews about WPV (worldwide)

no comprehensive literature research

Afolabi, A. A., Ilesanmi, O. S., & Chirico, F. (2024). Prevalence, pattern and factors associated with workplace violence against healthcare workers in Nigeria: A systematic review. Ibom Medical Journal, 17(2), 166-175.

Ayalew, E., Workineh, Y., Semachew, A., Woldgiorgies, T., Kerie, S., Gedamu, H., & Zeleke, B. (2021). Nurses' intention to leave their job in sub-Saharan Africa: A systematic review and meta-analysis. Heliyon, 7(6).

[-> Highest value in East Africa. That could be a consequence of WPV.]

Ayyash¹, S., Ruziqat, E., Alsmadi, A., Al Melhem¹, A., Eshah, N., Khalifeh, A., & Al Helou, D. (2023, September). Check for updates Violence Against Health Care Workers in Health Care Services: A Literature Review. In Proceedings of the Second International Nursing Conference" Nursing Profession in the Current Era"(INC 2023) (Vol. 67, p. 182). Springer Nature.

Ekpor, E., Kobiah, E., & Akyirem, S. (2024). Prevalence and predictors of workplace violence against nurses in Africa: A systematic review and meta‐analysis. Health Science Reports, 7(4), e2068.

[cited in your discussion]

Edward, K. L., Ousey, K., Warelow, P., & Lui, S. (2014). Nursing and aggression in the workplace: a systematic review. British journal of nursing, 23(12), 653-659.

Haleem, S. E. A. A., El Bingawi, H. M., Haleem, S. A., & El Bingawi, H. (2024). An international review of workplace violence against healthcare providers: Sudan as a case study. Cureus, 16(1).

Hallett, N., Gayton, A., Dickenson, R., Franckel, M., & Dickens, G. L. (2023). Student nurses' experiences of workplace violence: A mixed methods systematic review and meta-analysis. Nurse education today, 128, 105845.

Karatuna, I., Jönsson, S., & Muhonen, T. (2020). Workplace bullying in the nursing profession: A cross-cultural scoping review. International journal of nursing studies, 111, 103628.

Li, Y. L., Li, R. Q., Qiu, D., & Xiao, S. Y. (2020). Prevalence of workplace physical violence against health care professionals by patients and visitors: a systematic review and meta-analysis. International journal of environmental research and public health, 17(1), 299.

Liu, J., Gan, Y., Jiang, H., Li, L., Dwyer, R., Lu, K., ... & Lu, Z. (2019). Prevalence of workplace violence against healthcare workers: a systematic review and meta-analysis. Occupational and environmental medicine, 76(12), 927-937.

Lu, L., Dong, M., Wang, S. B., Zhang, L., Ng, C. H., Ungvari, G. S., ... & Xiang, Y. T. (2020). Prevalence of workplace violence against health-care professionals in China: a comprehensive meta-analysis of observational surveys. Trauma, Violence, & Abuse, 21(3), 498-509.

Nelson, S., Ayaz, B., Baumann, A. L., & Dozois, G. (2024). A gender-based review of workplace violence amongst the global health workforce—A scoping review of the literature. PLOS global public health, 4(7), e0003336.

Njaka, S., Edeogu, O. C., Oko, C. C., Goni, M. D., & Nkadi, N. (2020). Work place violence (WPV) against healthcare workers in Africa: A systematic review. Heliyon, 6(9).

Spector, P. E., Zhou, Z. E., & Che, X. X. (2014). Nurse exposure to physical and nonphysical violence, bullying, and sexual harassment: a quantitative review. International journal of nursing studies, 51(1), 72-84.

Tee, S., Özçetin, Y. S. Ü., & Russell-Westhead, M. (2016). Workplace violence experienced by nursing students: A UK survey. Nurse education today, 41, 30-35.

Worke, M. D., Koricha, Z. B., & Debelew, G. T. (2020). Prevalence of sexual violence in Ethiopian workplaces: systematic review and meta-analysis. Reproductive health, 17, 1-15.

[in relation to WPV against women]

Yusoff, H. M., Ahmad, H., Ismail, H., Reffin, N., Chan, D., Kusnin, F., ... & Rahman, M. A. (2023). Contemporary evidence of workplace violence against the primary healthcare workforce worldwide: a systematic review. Human resources for health, 21(1), 82.

Reviewer #4: Dear editor,

Thank you for providing me with the opportunity to review this interesting work. The authors present a systematic review and meta-analysis of the prevalence of workplace violence among healthcare workers in East Africa and of factors that moderate its prevalence. The topic is certainly of interest, but I have strong reservations against the paper in its present form. I address only the most pressing points for now. Further issues would need to be discussed after any revision.

1) The text is sometimes difficult to understand. I suggest the authors use ChatGPT or similar to improve the grammar and readability of their manuscript.

2) Out of 127 publications selected for full text review, 89 (i.e., about 75%) could not be accessed. I fully understand that access to research papers is limited at some institutions. However, a review that misses about three quarters of the relevant literature is not very useful. I suggest that the authors look for a collaboration partner who can provide them with access to the missing papers or find some other means to access these publications.

3) It remains unclear how the prevalence of workplace violence was assessed across studies. It is therefore unclear if the results can be pooled or compared. Imagine two studies that find 50% workplace violence. One study asked: “Did you experience workplace violence throughout the past 12 months?". The other asked “Have you heard of a colleague who experienced workplace violence throughout their career?” It would be misleading to assume that both studies found the same result. Please provide information how the experience of workplace violence was assessed in each study.

I wish the authors all the best for their project and hope that they further build on the substantial work they have undertaken so far.

Johannes Hönekopp

**Do you want your identity to be public for this peer review?** For information about this choice, including consent withdrawal, please see our Privacy Policy

Reviewer #1: No

Reviewer #2: No

Reviewer #3: No

Reviewer #4: **Yes: ** Johannes Hönekopp

---

## [Author Response · Author response to Decision Letter 1]

23 Jun 2025

Point by point response for editor and reviewers

Journal Name: PLOS ONE, Manuscript Number: PONE-D-24-46926

To editor

We would like to express our gratitude for the opportunity to submit our revised manuscript, and invaluable comments and the time and effort he invested for our work

Point by point response to Reviewer 1

Dear reviewer, thank you for the time and efforts you have dedicated for reviewing and your valuable suggestions to improve our work

Reviewer #1: Manuscript Number: PONE-D-24-46926

Manuscript Title: Magnitude of workplace violence and associated factors among health professionals in East Africa. Systematic review and meta-analysis

Congratulations dear authors on your scholarly work based on a priori protocol registered in PROSPERO; you have brought an important study problem with good findings that have public health importance in optimizing health care professional safety and well-being in work place. However, there few methodological issues that I want you to address before considering the manuscript for publication.

Reviewers’ questions or comments

Comments

Abstract

Introduction: This section needs to tell the reader about overview of burden of the problem, gap justification/ reason/aim of study.

Methods: better to remove the aim of the study from this part and add to introduction section. Please add sub group analysis, sensitivity analysis, study period etc…

Result: you need to reduce, make it short and precise (instead of writing associated variable and AOR separately write like (Emergency department (95%CI: 3.22, 5.39; I2 =93.71%, p = 0.00).Please write total study participants and included studies.

Conclusion: “The magnitude of workplace violence in the region was high” do you have benchmark to say high?

Your conclusions need to be based on your finding. I recommend you to place your Prospero registration here (Abstract)

Authors’ responses: Thank you for your comments and recommendations on improving structured abstract. 1. Introduction section: we have revised the content of the introduction section of the abstract based on your comments. 2. Methods section: in response to your recommendations, we have appropriately placed the study’s aim and added subgroup analysis, sensitivity analysis, and the study period in this sections of the abstract. 3. Results section: we have accept your suggestions to make the results section shorter and more precise, and we revised it accordingly. Additionally, we have added the total number of study participants and the number of included studies. 4. Conclusion and use of benchmark: Regarding the phrasing in the conclusion and the mention of a benchmark to define what is considered “high”, we acknowledge that there is no established benchmark in this region. Therefore, we have revised the expression to reflect this and updated it in the revised manuscript. Finally, the Prospero registration number has been added to the abstract, as per your recommendation. Please see page 2, for all corrections.

Reviewers’ questions or comments

Introduction (main): I feel it is well stated and organized except first write the full term before abbreviation, see your grammatical and language usages. In addition, you better to state the known factors/ variables, the attempted solution taken and planed strategies.

Authors’ responses: Thank you for your insightful comments and recommendations regarding the use of abbreviations. We have taken your feedback into account and conducted a thorough review of abbreviations, grammar, and overall language usage in the revised manuscript. In addition, we have incorporated your suggestions regarding inclusion of variables/factors identified previously in the introduction section. Please see page 3, lines 20-21.

Reviewers’ questions or comments

Methods: do you think that PIOC mnemonic is suitable for this review? Please revise it.

Please operationalize “workplace violence”

Authors’ responses: Thank you for your constructive comments regarding PIOC and the need for operational definitions of workplace violence. We have taken your recommendations into account and changed PIOC to CoCoP design, both in response to your recommendations and to better align with the aim of the study in the revised manuscript. In addition, we have clearly operationalized the concept of workplace violence. Please see page 4 and 5 for CoCoP and page 6 for operational definition.

Reviewers’ questions or comments

Result: Under “Characteristics of the included studies” be consistent write all in words or number (4, eighteen, 7)

It is better to narrate your result in intelligible, short and conclusive manner (revise it)

Authors’ responses: Thank you for your comments regarding the inconsistency use of words and numbers in the descriptions of the characteristics of the included studies, as well as your suggestions to present the results in a more intelligible, concise, and conclusive manner. We have taken your feedback into account and consistently used words throughout this section. The manuscript has been carefully revised to improve clarity and readability. Please see page 7, lines 8-12.

Reviewers’ questions or comments

Discussion: I recommend you to revise your discussion section critically.

• Please use reference for your justification,

• Make it short and clear to transfer your message well to the reader

• There are some sentences which placed unwisely like “As primary articles revealed that……?

• The flow of idea is not smart

• Please use paragraph for distinct idea

• Generally it needs deep revision

Authors’ responses: Thank you for insightful comments and recommendations regarding discussion section. We have carefully considered your suggestions and incorporated them into the revised manuscript. Specifically, we have provided appropriate references for our justifications, used shorter sentences to enhance clarity, removed or revised unclear statements, improved the logical flow of ideas, and structured the text into distinct paragraphs to better separate different points. Please refer to the discussion section (9, 10 and 11) of the revised manuscript. We look forward to resolving any remaining issues.

Reviewers’ questions or comments

Your conclusion should base your result

Authors’ responses: Thank you for your constructive feedback. We have revised the conclusion section based on your recommendations in the updated manuscript. Please see page 11, lines 10-18.

Reviewers’ questions or comments

General comments

There are several typological and grammar usage errors that need extensive proof reading for revisions.

Don’t use the conjunction ‘or’ to illustrate more, just one phrase or sentence is enough

Good luck!!

Authors’ responses: Thank you for your thorough review and for recommending that we proofread the manuscript to improve its readability. We have carefully proofread the revised manuscript. In addition, we have removed unnecessary conjunctions to enhance clarity and flow. Please see the revised manuscript. We look forward to improving any typological and grammatical errors.

Point by point response to Reviewer 2

Dear reviewer, thank you for the time and efforts you have dedicated for reviewing and your valuable suggestions to improve our work.

Reviewer #2: Thank you for the opportunity to review this manuscript. This meta-analysis estimates the prevalence of life-long workplace violence among healthcare workers in East Africa, understanding workplace violence as having suffered at least once an aggression from a client or user. The meta-analysis is generally well perfomed and contains thorough analyses. I have one major comment which I hope will help improve this manuscript.

Reviewers’ questions or comments

The meta-analysis 25 studies in its main analysis and obtains a robust estimation of the workplace violence prevalence. However, when it comes to analyzing moderator variables, the manuscript's approach is suboptimal in the sense that it does not take all of the possible studies. For instance, let me take the example of the analysis of sex as a potential variable. The manuscript only take those studies where men and women were compared and use the OR as a size effect. However, it is also possible to take the proportion of men and women in every study and perform a meta-regression, using the prevalence as the effect size, and thus taking all of the studies instead of only some of them. Same logic could be applied to age, and years of experience, and perhaps other moderating variables as well.

Authors’ responses: Thank you for your insightful comments regarding conducting a meta-regression using the proportion of predictors (significantly associated factors) with workplace violence. We appreciate your constructive feedback on performing meta-regression in this manner when encountering high heterogeneity among the included studies. While we extracted data on the proportions of these independent variables, we did not conduct a meta-regressions, as there was no substantial variability among the included studies. We will use your recommendation to perform meta-regression for future works while facing variability.

Reviewers’ questions or comments

The manuscript also mentions that the autohrs searched on databases like PubMed, Google Scholar, and Google. Is this the complete list of databases?

Authors’ responses: Thank you for your questions regarding the amount of listed databases. In short, the response is “No”. We retrieved articles from Web of Science, Google Scholar, PubMed, and Google manual search as well as online University repositories. Although we searched multiple databases to retrieve relevant studies, none of them contained studies beyond those already included and mentioned in this manuscript. Please see page 6.

Reviewers’ questions or comments

As a minor comment, regarding abbreviations and their use, it is customary that, the first time a term appears, both the expanded term and its abbreviation are mentioned, and in subsequent apprearances only the abbreviation is used. I suggest the manuscript to abide to this custom. For instance, the first time the term appears is in the first sentence: there I would say "Workplace violence (WPV) is a violent...", and use only the abbreviation WPV afterwards. Same rule may be applied to the abbreviation AOR.

Authors’ responses: Thank you for your constructive comments regarding the use of abbreviations and their explanations. We have taken your feedback into account and made the necessary corrections in the revised manuscript. Please refer to the complete revised manuscript for these corrections.

Reviewers’ questions or comments

Last, in the last paragraph of the Methods, it is said that p-value "<0.05% was used". Do you mean a significance threshold of 0.05?

Authors’ responses: Thank you for your comments regarding the description of p. value < 0.05%. In short, the answer to your question is “No”. To determine the pooled significance of statistically associated factors with the outcome variable, we used a 95% confidence interval. We meant “P-value of Q test < 0.05”. Please see page 6, line 21.

Point by point response to Reviewer 3

Dear reviewer, thank you for the time and efforts you have dedicated for reviewing and your valuable suggestions to improve our work.

Reviewers’ questions or comments

Reviewer #3: In a meta-analysis, this paper deals with the questions of how high workplace violence is in East African countries and which predictors were found. It is a methodologically sound study. This and the results found are the strength of the work. However, I do see a some weaknesses in the work. Apart from a few points at the end, the introduction is good, but a chapter with related studies is missing. I am thinking in particular of systematic reviews and meta-analyses on the issues relating to East Africa, but also worldwide for comparison. For this reason, the discussion also suffers because there is no holistic reference to that existing literature. The methods section is complete in itself, but some more precise explanations are missing, and it could generally be more detailed. In my view, a very important question is why 70% of the studies were excluded due to unavailability before the full text review. It is very important to assess whether these could have an influence on the results.

I think this study is an important contribution to the documentation and prevention of workplace violence. I hope that you can revise this meta-analysis so that it can be published.

Authors’ responses: Thank you for your constructive comments and for carefully reviewing the study selection process. When we refer to “full texts”, we are specifically looking for studies that include comprehensive details on the study population, study areas, and their outcomes to our research. We apologize for any unclear phrasing or wording in the original manuscript, we have revised the expression accordingly in the revised manuscript. Therefore, by “full texts unavailable” in this context, we meant that one or more of the above parameters were not accessible. Please see the page 7, lines 1- 6.

Reviewers’ questions or comments

General comments

• In chapter 3 (methods), 3.1 Data analysis follows 2.4 Data extraction. I suppose it should be 2.5 Data analysis.

You have no chapter 3. After chapter 3. Methods chapter 4. Results follows, and the 4. Meta-Analysis.

• Please, let proofread your paper. There as some grammatical errors and stylistic issues.

Authors’ responses: Thank you for your constructive comments regarding the order of topics and sub-topics. We apologize for inconsistent use of these sections in the original manuscript. We addressed all the issues in the revised version. Please see the updated methods section of manuscript.

Reviewers’ questions or comments

Abstract

The abstract is comprehensible and contains the relevant information.

• There are mentioned findings of a WHO study on WPV that are not mentioned in the introduction.

• I recommend to mention all databases searched not “like PubMed, …”.

• Egger’s test (not egger’s test)

• AOR is used in the abstract. I would write out the abbreviation in the abstract and introduce the abbreviation at first use later in the paper.

• Do not reference to figures in the abstract (“fig.12”).

• The second part of the results (in the abstract) “Emergency room AOR was …” is not written in complete sentences.

Authors’ responses: Thank you for your insightful comments and suggestions, which have helped improve the clarity and readability of our manuscript.

1. Data in abstract but not in the introduction: we have checked it, but the mentioned figure (20%-30%) in the introduction section of the abstract also found in the main introduction section of the original manuscript. Please see page 3 lines 7 - 8.

2. Missed databases in the abstract: we have included all databases used to retrieve articles in the methods section of the abstract in the revised manuscript.

3. Regarding capitalization, abbreviations, and fig 12 referenced in the abstract: we have made appropriate corrections in the revised manuscript. Please see the abstract section of the revised manuscript.

Reviewers’ questions or comments

Introduction

I see some issues in the last section of the introduction.

You write that WPV is “predominantly dangerous in Africa, especially in East African countries”. Why is that the case. You should explain this.

“they are almost similar in resource, health staff and technology usage, infrastructures, health care practice and environment.” -> similar to which countries.

You write that there is no comprehensive study on WPV in East Africa and that despite the importance there are few information. But then you write “It is therefore feasible to determine the magnitude workplace violence and its predictors in the region level.”

That is not logic to me. If there are few studies, few information, how can you conduct a meta-analysis?

• There are phrases that are not easy to understand for me (e.g. “having almost similar healthcare institutions and challenges in the region of this”)

• Introduce the abbreviation of workplace violence at it’s first use; not further down in the introduction.

• “the rest employees”

---

## [Decision Letter · Decision Letter 1]

17 Jul 2025

Dear Dr. Eshetie,

Thank you for submitting your manuscript to PLOS ONE. After careful consideration, we feel that it has merit but does not fully meet PLOS ONE’s publication criteria as it currently stands. Therefore, we invite you to submit a revised version of the manuscript that addresses the points raised during the review process.

The reviewers have different opinions on this manuscript. The author is requested to make major revisions to the fourth reviewer's suggestion, because this manuscript does have the problems he raised. The author is asked to reasonably explain the third reviewer's recommendation, especially since he listed four articles to refute the author's argument.

We look forward to receiving your revised manuscript.

Kind regards,

De-Chih Lee, Ph.D.

Academic Editor

PLOS ONE

Journal Requirements:

Additional Editor Comments (if provided):

The reviewers have different opinions on this manuscript. The author is requested to make major revisions to the fourth reviewer's suggestion, because this manuscript does have the problems he raised. The author is asked to reasonably explain the third reviewer's recommendation, especially since he listed four articles to refute the author's argument.

Reviewers' comments:

Reviewer's Responses to Questions

**Comments to the Author**

Reviewer #2: All comments have been addressed

Reviewer #3: (No Response)

Reviewer #4: (No Response)

2. Is the manuscript technically sound, and do the data support the conclusions?

Reviewer #2: Yes

Reviewer #3: Partly

Reviewer #4: Partly

3. Has the statistical analysis been performed appropriately and rigorously?

Reviewer #2: Yes

Reviewer #3: Yes

Reviewer #4: No

4. Have the authors made all data underlying the findings in their manuscript fully available?

Reviewer #2: Yes

Reviewer #3: Yes

Reviewer #4: Yes

5. Is the manuscript presented in an intelligible fashion and written in standard English?

Reviewer #2: Yes

Reviewer #3: No

Reviewer #4: Yes

Reviewer #2: The manuscript considerably improved after this revision. I have no further comments. I congratulate the authors for their well done work.

Reviewer #3: The authors have incorporated most of the reviewers' suggestions for improvement or explained why they have not done so. The paper has improved significantly since the revision and might be ready for publication with minor revisions if two points are clarified.

In all, three points remain unclear to me:

1. Proofreading is still required.

Examples: The first sentence in the introduction is still incorrect, or rather, an error has been inserted (WPV occur instead of WPV occurs). Interestingly, the same sentence is better formulated in the abstract (even before the revision). ‘Towards the staff’ (abstract) or ‘toward the staffs’ (introduction) may be correct, but I would write ‘against employees’. There are other examples such as: ‘the impact of predictors on outcome variable’ (that is ‘variables’?). In the abstract, the list of moderators is not integrated into a correct sentence (‘... with the outcome variable. Emergency department ...’).

All this makes reading somewhat more difficult and leads to misunderstandings.

2. Studies with no full text

I understand the explanation for the exclusion of studies lacking “full texts” (still written this way on page 5), but I am still not convinced. I apologize for being so suspicious and persistent here. I might be a question of language, or expression that I don’t understand correctly. But I cannot exclude that your explanation is a way to save the paper for publication.

The sentence ‘During full-text review, 89 studies with no accessible full text were removed’ is a clear and easily understandable statement that says that 89 studies were not accessible. The new wording ‘During full-text review, 89 studies were excluded due to the full-text articles did not report the outcome of interest’. This is a completely different statement. You excluded the studies because they did not meet the inclusion criteria.

3. Most studies from Ethiopia

Of the 25 studies analysed, 18 (72%) are from Ethiopia. I wonder whether there are really no studies on WPV from other East African countries such as Tanzania, Mozambique or Madagascar. In the revised version, this weakness is mentioned in the discussion: ‘Therefore, based on the subgroup analysis, we can conclude that the source of the heterogeneity may be the inclusion of studies from Ethiopia, Rwanda and Sudan.’ I understand the sentence, although it is not a clear statement: ‘we can conclude that [...] may be’.

However, in my view, there are studies on the subject from other East African countries: e.g.

- Patricio et al. (2022): https://doi.org/10.1002/hpm.3506

- Shimoda et al. (2020): https://doi.org/10.1186/s12884-020-03256-5

- El Ghaziri et al. (2014): https://doi.org/10.1016/j.jana.2013.07.002

If it is indeed the case that most of the studies are from Ethiopia, then this requires a plausible explanation or in-depth discussion.

If studies were indeed excluded because they were not accessible, and if there are indeed further studies from other East African countries, then I would not publish the paper. It would be better to complete data collection, reanalyse, adapt the text and resubmit the paper.

I apologize again for my mistrust, and I recommend speaking with the editor.

Reviewer #4: I find the manuscript much improved, and I like the clear focus on applied consequences in the discussion. A number of problems still need to be addressed, though.

1. P7, “13 studies were also removed due to study design differences”. What does this mean? Is this related to the exclusion of studies due to poor quality mentioned earlier? Please clarify.

2. Regarding the proportion of staff experiencing violence, you present a test of publication bias (p7). I don’t think that the concept of publication bias applies here. Whichever proportion of violence a study finds, applying any test of statistical significance does not make sense. It is just a single estimate, and there is no benchmark against which it could be sensibly tested against. Consequently, there cannot be bias against statistically non-significant studies. I therefore suggest you drop the test of publication bias.

3. The sensitivity analysis (p7/8) is very difficult to understand; please rephrase.

4. Given the high heterogeneity observed, I understand the motivation to look for moderators. However, your sub-group analyses are not very convincing. To test whether there is time trend strikes me as a good idea. However, what you should do is include publication year (or even better: year of data collection, if available) as a moderator in the meta-analysis. You can then check if this moderator is statistically significant. This would indicate that reported violence systematically increased or decreased over the time period studied. Similarly, the comparison of countries fails to address if the level of violence differs significantly across countries. Only if that is the case should country be considered a moderator. Alternatively, you might simply drop Figures 7 and 8 and the respective text.

5. “Similarly, inadequate staffing was statistically associated with workplace violence” (p8): In that case the confidence interval should not include 1, but you report it as “95% CI: 0.90, 3.97” (p8). Note, if a result is statistically significant at .05 the 95% CI should exclude the value that indicates ‘no effect’, i.e. 1 for OR. Please address this contradiction. If staffing proves statistically non-significant, the discussion and conclusion need to change accordingly.

6. Table 2: It is essential that you indicate for each study, what measure of workplace violence was used (or that relevant information is missing). If we don’t know what the studies asked, we don’t know what the overall prevalence means. If the studies assessed the prevalence of workplace violence in different ways, this might have contributed to the large observed heterogeneity.

A minor point: It should be “Likewise” instead of “Like ways” on p10.

Johannes Hönekopp

**Do you want your identity to be public for this peer review?** For information about this choice, including consent withdrawal, please see our Privacy Policy

Reviewer #2: No

Reviewer #3: No

Reviewer #4: **Yes: ** Johannes Hönekopp

---

## [Author Response · Author response to Decision Letter 2]

29 Jul 2025

Point by point response for editor and reviewers

Journal Name: PLOS ONE, Manuscript Number: PONE-D-24-46926

To editor

We would like to express our sincere gratitude to the editor for the opportunity to resubmit our revised manuscript. We deeply appreciate the invaluable comments, thoughtful guidance, and the time and effort dedicated to improving the quality of our work.

Editors’ questions or comments

The author is asked to reasonably explain the third reviewer's recommendation, especially since he listed four articles to refute the author's argument.

Authors’ response: Thank you for providing the list of articles mentioned by reviewer. While the editor’s comments indicate that we should respond to four (4) studies, we were able to identify only three (3) articles cited in the reviewer’s text. This is a gentle reminder to clarify the discrepancy between the stated number of articles (4) and the actual number we found (3). We were unable to locate the fourth article in the text forwarded from Reviewer 3.

Point by point response to Reviewer 3

Dear reviewer, thank you for dedicating your time and effort to review our work. We sincerely appreciate your insightful comments and valuable suggestions, which have been instrumental in helping us improve the clarity and readability of the manuscript.

Reviewer #3: The authors have incorporated most of the reviewers' suggestions for improvement or explained why they have not done so. The paper has improved significantly since the revision and might be ready for publication with minor revisions if two points are clarified.

In all, three points remain unclear to me:

Reviewers’ questions or comments

1. Proofreading is still required.

Examples: The first sentence in the introduction is still incorrect, or rather, an error has been inserted (WPV occur instead of WPV occurs). Interestingly, the same sentence is better formulated in the abstract (even before the revision). ‘Towards the staff’ (abstract) or ‘toward the staffs’ (introduction) may be correct, but I would write ‘against employees’. There are other examples such as: ‘the impact of predictors on outcome variable’ (that is ‘variables’?). In the abstract, the list of moderators is not integrated into a correct sentence (‘... with the outcome variable. Emergency department ...’).

All this makes reading somewhat more difficult and leads to misunderstandings.

Authors’ response: Thank you for your constructive feedback. We appreciate your recommendation regarding the sentence clarity. We have taken your recommendation into account and proofread the manuscript. Kindly refer to page 2 and 3 for specific feedback.

Reviewers’ questions or comments

2. Studies with no full text

I understand the explanation for the exclusion of studies lacking “full texts” (still written this way on page 5), but I am still not convinced. I apologize for being so suspicious and persistent here. I might be a question of language, or expression that I don’t understand correctly. But I cannot exclude that your explanation is a way to save the paper for publication.

The sentence ‘During full-text review, 89 studies with no accessible full text were removed’ is a clear and easily understandable statement that says that 89 studies were not accessible. The new wording ‘During full-text review, 89 studies were excluded due to the full-text articles did not report the outcome of interest’. This is a completely different statement. You excluded the studies because they did not meet the inclusion criteria.

Authors’ response: Thank you for your follow-up regarding your previous question. We greatly appreciate your insightful comments, which have helped improve the clarity and readability of the manuscript. Please accept our sincere apologies for the confusion caused by the wording. The revised wording was intend to clarify the meaning of our original expression, not to defend the paper. You are welcome to verify, through your own searching, the presence or absence of any eligible studies that were excluded due to inaccessible full-texts. We look forward to receiving any suggestions regarding to the expression of this exclusion.

Reviewers’ questions or comments

3. Most studies from Ethiopia

Of the 25 studies analysed, 18 (72%) are from Ethiopia. I wonder whether there are really no studies on WPV from other East African countries such as Tanzania, Mozambique or Madagascar. In the revised version, this weakness is mentioned in the discussion: ‘Therefore, based on the subgroup analysis, we can conclude that the source of the heterogeneity may be the inclusion of studies from Ethiopia, Rwanda and Sudan.’ I understand the sentence, although it is not a clear statement: ‘we can conclude that [...] may be’.

However, in my view, there are studies on the subject from other East African countries: e.g.

- Patricio et al. (2022): https://doi.org/10.1002/hpm.3506

- Shimoda et al. (2020): https://doi.org/10.1186/s12884-020-03256-5

- El Ghaziri et al. (2014): https://doi.org/10.1016/j.jana.2013.07.002

If it is indeed the case that most of the studies are from Ethiopia, then this requires a plausible explanation or in-depth discussion.

If studies were indeed excluded because they were not accessible, and if there are indeed further studies from other East African countries, then I would not publish the paper. It would be better to complete data collection, reanalyse, adapt the text and resubmit the paper.

I apologize again for my mistrust, and I recommend speaking with the editor.

Authors’ response: Thank you again for your insightful feedback on our manuscript. We appreciate your effort and dedication in searching and listing related studies, even though they fall outside our inclusion criteria. Further details are provided below.

- Patricio et al. (2022): https://doi.org/10.1002/hpm.3506, which conducted in Mozambique. According to standard classification by the Africa Union, United Nation, and World Bank, Mozambique is considered part of Southern Africa. However, our review included only studies conducted in East Africa. Therefore, the study by Patricio et al. was excluded based on geographic criteria. Please refer to the study’s setting for more details.

- Shimoda et al. (2020): https://doi.org/10.1186/s12884-020-03256-5 conducted in Tanzania. The study assessed workplace violence perpetrated by nurses and midwives against women during childbirth. As the literature shows, violence typically involves two key participants: perpetrators, who commit the act, and victims, who suffer harm or the threat of harm. Accordingly, the study examined self-reported instances of violent acts committed by nurses and midwives during childbirth. In this context (Shimoda et al.), nurses and midwives are considered perpetrators, not victims.

Although Tanzania is part of our study area and healthcare professionals (nurses and midwives) constitute our target population, this systematic review includes only studies that report workplace violence against healthcare professionals by service seekers. Therefore, the study by Shimoda et al. was excluded due to a mismatch in the target population, as it focused on violence committed by, rather than against, healthcare workers. Kindly see the full document of the study.

- El Ghaziri et al. (2014): https://doi.org/10.1016/j.jana.2013.07.002 conducted their study in Sub-Saharan Africa, which broadly includes East, West, Central, and Southern regions. However, their sample included healthcare professionals from countries such as Nigeria, Zimbabwe, Zambia, and Togo —none of which are part of East Africa. Therefore, the study by El Ghaziri et al. was excluded due to differences in the study area’s geographic focus. Please refer to the full document for a complete list of included countries.

Reviewers’ questions or comments

If it is indeed the case that most of the studies are from Ethiopia, then this requires a plausible explanation or in-depth discussion.

Authors’ response: Thank you for your question regarding the dominance of studies from Ethiopia. Indeed, a large proportion of the included studies were conducted in Ethiopia, and have acknowledged this in the limitation section of the manuscript based on previous comments by reviewers. This dominance may reflect both the research focus and the availability of data on the topic in the region, rather than a deliberate selection bias.

Reviewers’ questions or comments

I understand the sentence, although it is not a clear statement: ‘we can conclude that [...] may be’.

Authors’ response: Thank you for your feedback regarding the ambiguity between expressions certainty (can conclude) and uncertainty (may be). We have taken your insightful feedback into account and revised the expression. Please see page 8, lines 19-24.

Point by point response to Reviewer 4

Dear reviewer, thank you for dedicating your time and effort to review our work. We sincerely appreciate your insightful comments and valuable suggestions, which have been instrumental in helping us improve the clarity and readability of the manuscript.

Reviewer #4: I find the manuscript much improved, and I like the clear focus on applied consequences in the discussion. A number of problems still need to be addressed, though.

Reviewers’ questions or comments

1. P7, “13 studies were also removed due to study design differences”. What does this mean? Is this related to the exclusion of studies due to poor quality mentioned earlier? Please clarify.

Authors’ response: Thank you for your feedback regarding the need for clarity on the study design differences. We appreciate your effort and dedication in helping us improve the clarity of manuscript.

Regarding the 13 studies that were removed due to study design differences: This means they were excluded specifically because their study designs did not align with our predefined inclusion criteria, not due to poor quality. Importantly, this exclusion occurred prior to the quality assessment step, which was conducted only on studies that fully met the inclusion criteria.

Reviewers’ questions or comments

2. Regarding the proportion of staff experiencing violence, you present a test of publication bias (p7). I don’t think that the concept of publication bias applies here. Whichever proportion of violence a study finds, applying any test of statistical significance does not make sense. It is just a single estimate, and there is no benchmark against which it could be sensibly tested against. Consequently, there cannot be bias against statistically non-significant studies. I therefore suggest you drop the test of publication bias.

Authors’ response: Thank you for your constructive comment regarding the test of publication bias. We really appreciate your suggestion and feedback on this test. In meta-analysis of proportion (prevalence) data, each study contributes a single effect estimate, which can be used to construct a funnel plot in STATA. Although each study reports only one estimate, a funnel plot can still be created by plotting the following:

X-axis (horizontal line): The effect size, typically the logit-transformed prevalence (proportion) from each study.

Y-axis (vertical line): A measure of precision, commonly the inverse of the standard error of the transformed estimate (prevalence).

This funnel plot is centered on the pooled estimate and includes 95% confidence intervals, allowing for visualization of the spread of effect sizes relative to their precision— even when based on a single estimate. In our study, the funnel plot used to assess publication bias was constructed according to this principle. The asymmetry observed in the funnel plot was confirmed by Egger’s test.

Egger’s test examines weather there is a systematic relationship between the effect sizes of the study and their standard errors (a proxy for study precision). Egger’s test was conducted by regressing the standard normal deviate (i.e., the transformed prevalence divided by its standard error) on study precision (defined as the inverse of the standard error). If smaller studies tend to report disproportionally higher effects, the regression intercept will deviate from zero. A statistically significant non-zero intercept (p <0.05) indicates the presences of small-study effects, which may reflect publication bias. Egger’s test quantifies this asymmetry and assesses whether it is statistically significant. For your information, the references we used are listed below.

1, Sterne JAC, Egger M. Funnel plots for detecting bias in meta-analysis: Guidelines on choice of axis. J Clin Epidemiol 2001; 54: 1046-1055. The study discussing how Funnel pots should be constructed, especially relevant when dealing with proportion or prevalence.

2, Barendregt JJ, Doi SA, Lee YY, Norman RE, Vos T. Meta-analysis of prevalence. J epidemiol community health. 2013 Nov 1;67(11):974-8. Propose the double arcsine transformation, which is essential when dealing with proportion in meta-analysis and helps when plotting funnel plot.

3, Higgins, J. P. T., Thomas, J., Chandler, J., et al. (Eds.). (2022), Cochrane Handbook for Systematic Reviews of Interventions: chapter 13 includes detailed discussion on reporting biases and the uses of funnel plots. It provides guidelines on interpreting funnel plot asymmetry and properly applying Egger’s regression test (requiring at least 10 studies, interpretation alongside visual inspection). We welcome any feedback in case we have misunderstood your question.

Reviewers’ questions or comments

3. The sensitivity analysis (p7/8) is very difficult to understand; please rephrase.

Authors’ response: Thank you for your comment regarding difficulty in understanding the sensitivity analysis section. We have taken your feedback into account and made the necessary corrections in the revised manuscript. Kindly refer to page 8, lines 5-8.

Reviewers’ questions or comments

4. Given the high heterogeneity observed, I understand the motivation to look for moderators. However, your sub-group analyses are not very convincing. To test whether there is time trend strikes me as a good idea. However, what you should do is include publication year (or even better: year of data collection, if available) as a moderator in the meta-analysis. You can then check if this moderator is statistically significant. This would indicate that reported violence systematically increased or decreased over the time period studied. Similarly, the comparison of countries fails to address if the level of violence differs significantly across countries. Only if that is the case should country be considered a moderator. Alternatively, you might simply drop Figures 7 and 8 and the respective text.

Authors’ response: Thank you for your insightful comments regarding the subgroup analyses and the subsequent meta-regression used to investigate the sources of the observed high heterogeneity.

Given the high heterogeneity observed in the overall meta-analysis, we conducted subgroup analyses based on publication year, average sample size, and study region. Subgroup analyses by publication year and sample size did not show a notable reduction in heterogeneity within groups. However, the analysis based on study region revealed a reduction in heterogeneity in some groups.

To statistically support our subgroup findings, we conducted meta-regression analyses for all comparisons. The I2 statistics from the subgroup analyses based on publication year and sample sizes were consistent with the meta-regression results (p = 0.26 for publication year and p= 0.42 for average sample size), suggesting these factors were not major sources of observed heterogeneity. In contrast, the subgroup analysis by study region showed a reduction in heterogeneity in some groups. However, the meta-regression result for this variable was not statistically significant (p = 0.39), which limits our ability to draw a definitive conclusion about the study region as a source of heterogeneity. This inconsistency may be due to the limited number of studies within each reginal group or other unmeasured confounding factors such as differences in health

---

## [Decision Letter · Decision Letter 2]

15 Aug 2025

Dear Dr. Eshetie,

Thank you for submitting your manuscript to PLOS ONE. After careful consideration, we feel that it has merit but does not fully meet PLOS ONE’s publication criteria as it currently stands. Therefore, we invite you to submit a revised version of the manuscript that addresses the points raised during the review process.

We look forward to receiving your revised manuscript.

Kind regards,

De-Chih Lee, Ph.D.

Academic Editor

PLOS ONE

Journal Requirements:

Additional Editor Comments (if provided):

There are still some minor problems with this manuscript. The authors are requested to make minor revisions based on the first reviewer's comments.

Reviewers' comments:

Reviewer's Responses to Questions

**Comments to the Author**

Reviewer #2: All comments have been addressed

Reviewer #3: All comments have been addressed

2. Is the manuscript technically sound, and do the data support the conclusions?

Reviewer #2: Yes

Reviewer #3: Yes

3. Has the statistical analysis been performed appropriately and rigorously?

Reviewer #2: Yes

Reviewer #3: Yes

4. Have the authors made all data underlying the findings in their manuscript fully available?

Reviewer #2: Yes

Reviewer #3: Yes

5. Is the manuscript presented in an intelligible fashion and written in standard English?

Reviewer #2: Yes

Reviewer #3: Yes

Reviewer #2: Thank you for the opportunity to review the new version of this manuscript. I have two minor comments from this new version and the responses provided to the reviewers.

1. The manuscript says that "this review included all quantitative and mixed-methods observational studies". This is an inclusion criterion. Later, explaining the included and excluded articles, the manuscript says that 13 studies were excluded because the design did not align with inclusion criteria. However, I understand that those 13 studies reported the desired outcome, which is a prevalence, and thus I also understand that those 13 studies were quantitative. Of couse, I do not know which studies were excluded for this reason, but I can not help but wonder how were those studies performed to have the desired prevalence and yet not being included.

2. I agree with a comment formerly made by another reviewer that meta-regression using year as the independent variable would be a good idea. First, perhaps re-ordering the studies in Figure 2 by year would allow us to see possible time trends in the prevalence of workplace agressions; I must admit I do not see how studies in Figure 2 are currently ordered and perhaps a different order could be more helpful, but I would not be very insistive on this. Second, and more importantly, the authors divided the studies in two groups: before 2022 and after 2022, but there is still much heterogeneity. I would suggest treating the year as a quantitative variable in the meta-regression, instead of dichotomizing it.

Reviewer #3: The current version of the paper reads better and has improved in quality once again. Conducting a meta-analysis is a time consuming and hard work - I am very aware of this.

There are a few minor improvements that I recommend below.

The authors have argued well why they had to exclude many studies. The explanations for the three studies I mentioned have shown me this in an exemplary manner.

I would like to thank the authors for their great patience in responding to my objections with the relevant explanations and apologise once again for my mistrust.

From the studies included in the meta-analysis and their composition in terms of the countries in which they were conducted, I conclude that there is a significant lack of high-quality studies on the topic of WPV in the healthcare sector in most East African countries. This could be a conclusion from your study.

Points for improvement:

- You write: "20% to 38% of healthcare workers have experienced physical violence at some point during their careers, compared to employees in other sectors."

This concerns this sentence in the abstract and a similar sentence in the introduction.

I am missing the comparison. I would add some values of WPV from other areas here.

- In the abstract, you list the associated factors three times in a row, once in the same sentence (with and without AOR values) and then again in the conclusion.

I recommend rephrasing the sentence and the conclusion.

- Result section: ‘An extensive search was conducted, and a total of 1,243 studies were retrieved from databases such as Google’

better: ‘An extensive search was conducted, and a total of 1,243 studies were retrieved from the databases Google, ...’

- Publication bias: ‘The p-value from Egger's regression test (p < 0.03), along with the asymmetric distribution of the included primary studies on the funnel plot (Fig 3a), and suggests the presence of publication bias.’

-> ‘The p-value from Egger's regression test (p < 0.03), along with the asymmetric distribution of the included primary studies on the funnel plot (Fig 3a), suggests the presence of publication bias.’

**Do you want your identity to be public for this peer review?** For information about this choice, including consent withdrawal, please see our Privacy Policy

Reviewer #2: No

Reviewer #3: No

---

## [Author Response · Author response to Decision Letter 3]

18 Aug 2025

Point by point response for editor and reviewers

Journal Name: PLOS ONE, Manuscript Number: PONE-D-24-46926R2

Title: Magnitude of workplace violence and associated factors among healthcare professionals in East Africa: A Systematic Review and Meta-Analysis.

To editor

We sincerely thank the editor for granting us the opportunity to resubmit our revised manuscript. We greatly appreciate the valuable comments, thoughtful guidance, and the time and effort invested in helping us improve the quality of our work.

Point by point response to Reviewer 2

Dear reviewer, we are truly grateful for the time and effort you dedicated to reviewing our manuscript. Your insightful comments and constructive suggestions have been invaluable in guiding our revisions and have significantly contributed to improving the clarity, coherence, and overall quality of the paper. We sincerely appreciate your thoughtful feedback and the role it has played in strengthening our work

Reviewer #2: Thank you for the opportunity to review the new version of this manuscript. I have two minor comments from this new version and the responses provided to the reviewers

Reviewers’ questions or comments

1,The manuscript says that "this review included all quantitative and mixed-methods observational studies". This is an inclusion criterion. Later, explaining the included and excluded articles, the manuscript says that 13 studies were excluded because the design did not align with inclusion criteria. However, I understand that those 13 studies reported the desired outcome, which is a prevalence, and thus I also understand that those 13 studies were quantitative. Of couse, I do not know which studies were excluded for this reason, but I can not help but wonder how were those studies performed to have the desired prevalence and yet not being included.

Authors’ response: Thank you for your comments regarding the study selection process, specifically the exclusion of the 13 studies.

The primary and secondary outcomes of interest in this study are the prevalence of workplace violence and its associated factors, respectively. Accordingly, we included studies that report either prevalence and/or associated factors of workplace violence. Although systematic reviews, meta-analyses, clinical trials, case studies, and qualitative studies can report these outcomes, they were excluded due to differences in study design. This explains the exclusion of the 13 studies. However, reporting the outcomes of interest does not necessarily imply that the studies exclusively quantitative, as qualitative studies can also report associated factors for violence. For example, see the study, “Workplace Violence against Nurses in Uganda’s Public Hospitals: A Phenomenological Study” (available at: Makerere University Repository). Please use the following link to access it. https://makir.mak.ac.ug/handle/10570/14186. The most appropriate exclusion criterion for this study is the study design, rather than the study area, study population, or outcome of interest. We welcome any suggestion for further improving the clarity of this expression.

Reviewers’ questions or comments

2, I agree with a comment formerly made by another reviewer that meta-regression using year as the independent variable would be a good idea. First, perhaps re-ordering the studies in Figure 2 by year would allow us to see possible time trends in the prevalence of workplace agressions; I must admit I do not see how studies in Figure 2 are currently ordered and perhaps a different order could be more helpful, but I would not be very insistive on this. Second, and more importantly, the authors divided the studies in two groups: before 2022 and after 2022, but there is still much heterogeneity. I would suggest treating the year as a quantitative variable in the meta-regression, instead of dichotomizing it.

Authors’ response: Thank you for your valuable comments regarding the reordering of figure 2 and the use of year as moderator in the meta-regression analysis to explore the source of variability among the included primary articles.

1, Figure 2 presents the pooled prevalence of workplace violence, the individual study estimates, heterogeneity, and the study weights. In fact, we did not consider the publication year, the alphabetic order of the first author’s name, or any other characteristics when constructing the forest plot of the pooled prevalence of workplace violence. For your information the current order is based on the study region. There was no specific intention behind this ordering—it simply reflects the sequence of data collection. Arranging studies chronologically by year of publication could allow readers to visually perceive weather prevalence is increasing or decreasing over time; however, this would only provide a rough visual impression, not a formal statistical test of trend. To determine whether the prevalence of workplace violence significantly changes over time, a meta-regression analysis would be required. We are willing to adjust the ordering of figure 2 if deemed necessary.

2, Given the heterogeneity among the included articles, we employed several methods to explore its potential sources. One approach was subgroup analysis based on year of publication, however, this did not reduce heterogeneity within the groups. This may be due to the small number of studies in each group or other unmeasured factors. We also conducted a meta-regression using year of publication as moderator, but the result was not statistically significant (p = 0.26). For details please see page 8, lines 10 - 24.

Point by point response to Reviewer 3

Dear reviewer, we are sincerely grateful for the time and effort you dedicated to reviewing our manuscript. Your insightful comments and constructive suggestions have been invaluable in guiding our revisions and have greatly contributed to enhancing the clarity, coherence, and overall quality of the paper. We deeply appreciate your thoughtful feedback and the role it has played in strengthening our work

Reviewer #3: The current version of the paper reads better and has improved in quality once again. Conducting a meta-analysis is a time consuming and hard work - I am very aware of this.

There are a few minor improvements that I recommend below.

The authors have argued well why they had to exclude many studies. The explanations for the three studies I mentioned have shown me this in an exemplary manner.

I would like to thank the authors for their great patience in responding to my objections with the relevant explanations and apologise once again for my mistrust.

From the studies included in the meta-analysis and their composition in terms of the countries in which they were conducted, I conclude that there is a significant lack of high-quality studies on the topic of WPV in the healthcare sector in most East African countries. This could be a conclusion from your study.

Reviewers’ questions or comments

Points for improvement:

- You write: "20% to 38% of healthcare workers have experienced physical violence at some point during their careers, compared to employees in other sectors."

This concerns this sentence in the abstract and a similar sentence in the introduction.

I am missing the comparison. I would add some values of WPV from other areas here.

Authors’ response: Thank you for your comment regarding the need to add a numeric value for comparison. We have taken your feedback into account and incorporated the numeric value accordingly. Please see page 3, line 11.

Reviewers’ questions or comments

- In the abstract, you list the associated factors three times in a row, once in the same sentence (with and without AOR values) and then again in the conclusion.

I recommend rephrasing the sentence and the conclusion.

Authors’ response: Thank you for your insightful comments regarding the unnecessary repetition of similar expressions about associated factors. We have accepted your constructive feedback and made the necessary corrections accordingly. Kindly refer to page 2 of revised manuscript.

Reviewers’ questions or comments

- Result section: ‘An extensive search was conducted, and a total of 1,243 studies were retrieved from databases such as Google’

better: ‘An extensive search was conducted, and a total of 1,243 studies were retrieved from the databases Google, ...’

Authors’ response: Thank you for your suggestion regarding improving the sentence structure. We have taken your comment into account and incorporated it into the revised manuscript. Please see page 7, line 4.

Reviewers’ questions or comments

- Publication bias: ‘The p-value from Egger's regression test (p < 0.03), along with the asymmetric distribution of the included primary studies on the funnel plot (Fig 3a), and suggests the presence of publication bias.’

-> ‘The p-value from Egger's regression test (p < 0.03), along with the asymmetric distribution of the included primary studies on the funnel plot (Fig 3a), suggests the presence of publication bias.’

Authors’ response: Thank you for your valuable comment regarding sentence clarity. We have corrected in the revised manuscript. Kindly refer to page 7, line 28.

---

## [Decision Letter · Decision Letter 3]

31 Aug 2025

Magnitude of workplace violence and associated factors among healthcare professionals in East Africa: A Systematic Review and Meta-Analysis.     .

PONE-D-24-46926R3

Dear Dr. Eshetie,

A few minor errors (see review comments) need to be revised.

We’re pleased to inform you that your manuscript has been judged scientifically suitable for publication and will be formally accepted for publication once it meets all outstanding technical requirements.

Kind regards,

De-Chih Lee, Ph.D.

Academic Editor

PLOS ONE

Additional Editor Comments (optional):

Reviewer #2:

Reviewer #3:

Reviewers' comments:

Reviewer's Responses to Questions

**Comments to the Author**

Reviewer #2: All comments have been addressed

Reviewer #3: All comments have been addressed

2. Is the manuscript technically sound, and do the data support the conclusions?

Reviewer #2: Yes

Reviewer #3: Yes

3. Has the statistical analysis been performed appropriately and rigorously?

Reviewer #2: Yes

Reviewer #3: Yes

4. Have the authors made all data underlying the findings in their manuscript fully available?

Reviewer #2: Yes

Reviewer #3: Yes

5. Is the manuscript presented in an intelligible fashion and written in standard English?

Reviewer #2: Yes

Reviewer #3: Yes

Reviewer #2: (No Response)

Reviewer #3: Thank you for accepting my propositions.

The paper again improved.

I found only some tipos:

- page 10, line 14: all age group -> all age groups

- page 13, paper 6: the link contains spaces and misses a character ("-"; violence-against)

https://www.who.int/activities/preventing-violenceagainst-health-workers

-> https://www.who.int/activities/preventing-violence-against-health-workers

**Do you want your identity to be public for this peer review?** For information about this choice, including consent withdrawal, please see our Privacy Policy

Reviewer #2: No

Reviewer #3: No

---

## [Editor Report · Acceptance letter]

PONE-D-24-46926R3

PLOS ONE

Dear Dr. Eshetie,

I'm pleased to inform you that your manuscript has been deemed suitable for publication in PLOS ONE. Congratulations! Your manuscript is now being handed over to our production team.

Kind regards,

on behalf of

Dr. De-Chih Lee

Academic Editor

PLOS ONE